EMBO
*reports*

# Appropriate glycemic management protects the germline but not the uterine environment in hyperglycemia

Allan Zhao [1], Hong Jiang[1,7], Arturo Reyes Palomares [2,7], Alice Larsson [1], Wenteng He [1], Jacob Grünler[3], Xiaowei Zheng[3], Kenny A Rodriguez Wallberg [2,4], Sergiu-Bogdan Catrina [3,5] & Qiaolin Deng [1,6 ✉]

## Abstract

**Emerging evidence indicates that parental diseases can impact the health of subsequent generations through epigenetic inheritance. Recently, it was shown that maternal diabetes alters the metaphase II oocyte transcriptome, causing metabolic dysfunction in offspring. However, type 1 diabetes (T1D) mouse models frequently utilized in previous studies may be subject to several confounding factors due to severe hyperglycemia. This limits clinical translatability given improvements in glycemic control for T1D subjects. Here, we optimize a T1D mouse model to investigate the effects of appropriately managed maternal glycemic levels on oocytes and intrauterine development. We show that diabetic mice with appropriate glycemic control exhibit better long-term health, including maintenance of the oocyte transcriptome and chromatin accessibility. We further show that human oocytes undergoing in vitro maturation challenged with mildly increased levels of glucose, reflecting appropriate glycemic management, also retain their transcriptome. However, fetal growth and placental function are affected in mice despite appropriate glycemic control, suggesting the uterine environment rather than the germline as a pathological factor in developmental programming in appropriately managed diabetes.**

**Keywords** Diabetes; Placenta; Developmental Programming; Hypoxia; Fetal Development
**Subject Categories** Development; Metabolism

## Introduction

The prevalence of obesity and metabolic disease is rapidly increasing and emerging as a major global public health issue as a result of complex interactions between genetic predisposition and environmental factors. Epigenetic inheritance driven by environmental factors is a concept that elucidates how environmental exposures can induce heritable changes in gene expression patterns and phenotypic outcomes across generations mediated through various epigenetic modifications (Heard and Martienssen, 2014). The germline and uterine environment are acknowledged as independent maternal factors that can contribute to the process of epigenetic inheritance and result in alterations of phenotypic traits in adult offspring (Fitz-James and Cavalli, 2022). More and more studies have revealed that maternal exposure to environmental factors such as obesity, endocrine disruptors and excessive androgen plays a significant role in shaping the reproductive and metabolic health outcomes of adult offspring (Huypens et al, 2016; Han et al, 2018; Risal et al, 2019; Chen et al, 2022). However, the distinct contributions of germline versus uterine environment in shaping developmental outcomes within disease contexts remain inadequately investigated and understood. In vitro fertilization followed by fostering with healthy surrogate females has been utilized to exclude the effects of the uterine environment and ensure epigenetic inheritance via the gametes in mice. Through this strategy, it has been shown that gametes from high-fat diet induced obese females produce offspring more prone to develop obesity and diabetes, especially in males (Huypens et al, 2016; Han et al, 2018). With a similar approach, a recent study showed that maternal hyperglycemia in a severe type 1 diabetes (T1D) mouse model causes epigenetic inheritance of glucose intolerance in offspring due to significant alterations of the mature or Metaphase II (MII) oocyte transcriptome (Chen et al, 2022). These findings indicate a significant compromise in the germline integrity even prior to gestation, raising concerning implications for subjects with T1D.

[1]Department of Physiology and Pharmacology, Karolinska Institutet, Stockholm, Sweden. [2]Department of Oncology-Pathology, Karolinska Institutet, Stockholm, Sweden. [3]Department of Molecular Medicine and Surgery, Karolinska Institutet, Stockholm, Sweden. [4]Division of Gynecology and Reproduction, Department of Reproductive Medicine, Karolinska University Hospital, Stockholm, Sweden. [5]Center for Diabetes, Academic Specialist Centrum, Stockholm, Sweden. [6]Center for Molecular Medicine, Karolinska University Hospital, Stockholm, Sweden. [7]These authors contributed equally: Hong Jiang, Arturo Reyes Palomares. ✉E-mail: qiaolin.deng@ki.se

Therefore, it is imperative to revisit these results and confirm their validity and impact.

T1D is a prevalent chronic disease with a consistent rise in incidence observed worldwide in recent decades (Patterson et al, 2009; Berhan et al, 2011; Gregory et al, 2022). Given the early onset of T1D, a significant number of patients will suffer from the condition along their reproductive age, potentially impacting both the germline function and the pregnancy. Women with T1D have traditionally experienced reproductive abnormalities and fertility issues, owing to poor disease management and glycemic control, and one goal in the St. Vincent Declaration from 1989 is to improve fertility and pregnancy outcomes for women with diabetes (Thong et al, 2020; Diabetes care and research in Europe: the Saint Vincent declaration, 1990). However, advances in therapeutic options and stricter glycemic control have significantly enhanced the fertility in women with T1D (Jonasson et al, 2007). New technological advances such as continuous glucose monitoring (CGM) devices (Beck et al, 2017; Lind et al, 2017; Šoupal et al, 2020) and hybrid closed-loop systems, which has been successfully tested in many age groups including children (Breton et al, 2020), adolescents and young adults (Bergenstal et al, 2021) has further advanced and improved glycemic control. These clinical advances raise questions about the translatability of the recent finding of maternal inheritance of glucose intolerance via oocyte TET3 insufficiency. The T1D mouse model used has a severe metabolic phenotype and exhibits poor fertility and anovulation, characteristics which are now less clinically relevant (Chen et al, 2022; Lee et al, 2019). There is therefore a need to investigate how appropriately managed T1D affects the germline, which is currently not known.

Notably, despite the improved general health of T1D patients, several epidemiological studies have revealed that even with appropriate glycemic control, defined as HbA1c within target levels (<6.5% or <48 mmol/mol), T1D pregnancies still carry a higher risk of adverse outcomes, including birth defects, preterm birth, and more importantly fetal growth deviations (Ludvigsson et al, 2018, 2019). Moreover, the link between maternal T1D and offspring metabolic dysfunction was recently shown to be unrelated to glycemic control during pregnancy (Vlachová et al, 2015). As fetal growth deviations at birth are highly associated with development of metabolic disease later in life (Whincup et al, 2008; Knop et al, 2018; Wei et al, 2003), these results collectively imply persisting epigenetic inheritance of metabolic disease originating from the pregestational or gestational period even in appropriately managed T1D. However, the previous knowledge about appropriately managed T1D and pregnancy is based on epidemiological and clinical data. Hence, the specific cause of these adverse pregnancy outcomes and what molecular changes are associated with appropriately managed T1D remain unclear. It is also unknown whether these issues arise from a dysregulated germline or a disturbed uterine environment. Further research is therefore imperative to identify potential targets for prevention and to create insight into the specific roles of pregestational and gestational exposure in appropriately managed T1D.

Hence, in this study we sought out to investigate if it is the germline or the uterine environment that drives the potential epigenetic inheritance of metabolic disease in appropriately managed T1D. We first observed that the number of female subjects with T1D in reproductive age with appropriate glycemic control is rapidly increasing, which confirms the efficiency of recent therapeutic advances and highlights the urgency to study this patient group. Thereafter we investigated if glycemic levels corresponding to appropriately managed T1D, better reflecting the pathophysiology of contemporary T1D patients, will significantly affect the oocyte transcriptome. To do so, we optimized a maternal T1D mouse model using streptozotocin (STZ) given in low doses on five consecutive days. Different from previously widely used models, our female mice exhibited normal overall health, including adequate reproductive function and body weight gain. Subsequently, we employed superovulation techniques on females to retrieve MII oocytes for single-cell multi-omic sequencing using Smart3-ATAC. We observed no discernible differences in either gene expression or chromatin accessibility between oocytes derived from females treated with STZ and those from control females. Moreover, when challenged with mildly increased glucose concentrations, human oocytes undergoing in vitro maturation still retained their transcriptomic signatures. Interestingly, fetuses derived from our mouse model displayed significantly altered fetal growth coupled with placental dysfunction. Collectively, our results provide the first evidence regarding the crucial role of glycemic control and disease management in preserving the integrity of the oocyte transcriptome in T1D. Instead, our findings pinpoint the uterine environment as the pathogenic factor underlying fetal growth deviation in glycemic levels corresponding to appropriately managed T1D, potentially enhancing offspring future disease susceptibility.

## Results

Several recent randomized controlled trials show the benefit of recent therapeutic advances in T1D care (Beck et al, 2017; Lind et al, 2017; Šoupal et al, 2020; Breton et al, 2020; Bergenstal et al, 2021). To confirm that these therapeutic advances indeed yields improved glycemic control on a population level, we investigated the trends in glycemic care for women with T1D of reproductive age (18–45 years) over the last five years in Sweden using data from the Swedish National Diabetes Registry (Nationella Diabetesregistret). Indeed, the percentage of female subjects with glycated hemoglobin (HbA1c) levels <52 mmol/mol (counted as appropriate glycemic control) increased from 25.1% to 41.0% between the period of 2017–2022 nationwide (Fig. 1A). During the same period, the percentage of patients with HbA1c levels >70 mmol/mol (counted as poor glycemic control) decreased from 20.4% to 12.7% (Fig. 1A). These results confirm continuous improvements in glycemic management for all female subjects with T1D of reproductive age, further emphasizing the importance to model this understudied group of patients and study the impact of improved glycemic management on the germline integrity and uterine environment.

Notably, T1D mouse models used to this date mostly develop severe diabetes with poor glycemic control and general health status. Indeed, a previous study investigating the effects of maternal T1D on the oocytes used severely diabetic mouse models with extreme blood glucose levels, generated by one STZ injection at a high dose (i.e., 150 mg/kg) (Chen et al, 2022). As the therapeutic options have steadily improved, the translational relevance of these severely diabetic mouse models is questionable. Although the most consensus factor underlying the development of diabetes

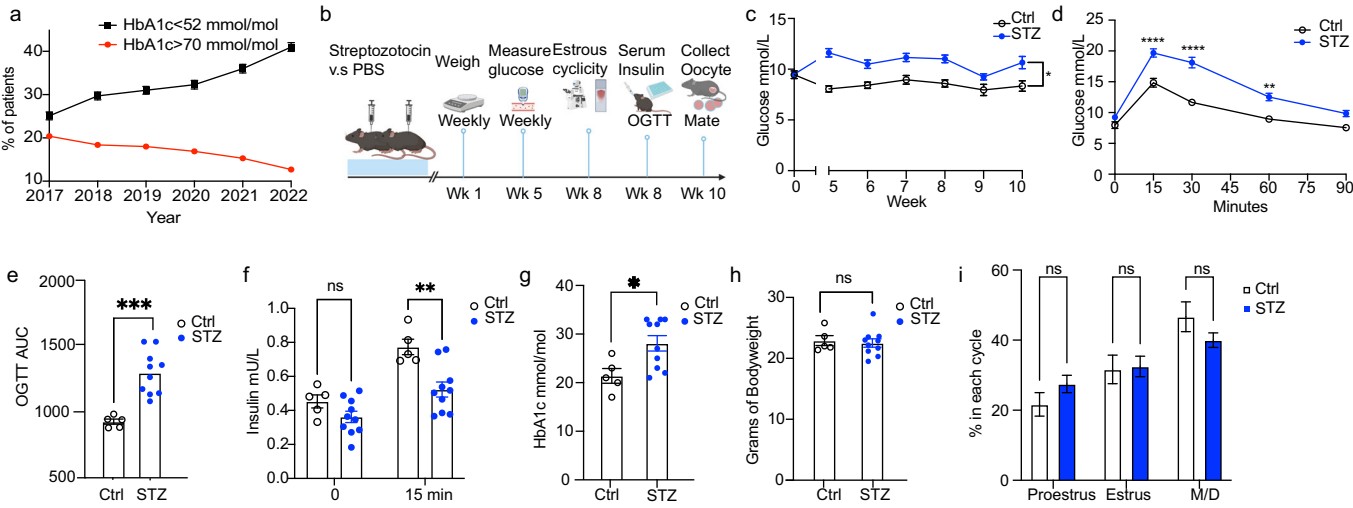

**Figure 1. Phenotyping of a model of appropriately managed type 1 diabetes.**

(A) Trends in glycemic control for all women of reproductive age (18–45 years of age, $n = 9771$ 2017, $n = 9951$ 2018, $n = 9983$ 2019, $n = 9536$ 2020, $n = 9964$ 2021, $n = 10129$ 2022) with a type 1 diabetes diagnosis in Sweden between 2017 and 2022. Red color indicates percentage (%) of patients with bad glycemic control (HbA1c > 70 mmol/mol). Black color indicates percentage (%) of patients with good glycemic control (HbA1c < 52 mmol/mol). (B) Schematic for mouse experiment of optimized STZ model. (C) Trends of morning blood glucose levels in Control and STZ groups over the time course of the experiment. (D) Blood glucose levels at different timepoints during an oral glucose tolerance test (OGTT). (E) Glucose area under the curve (AUC) from the OGTT experiment in (C). (F) Serum insulin levels at 0 and 15 min timepoints from OGTT experiment in (B). (G) HbA1c levels 10 weeks after induction. (H) Body weight of mice 10 weeks after STZ exposure. (I) Time spent in every phase of estrous cyclicity. STZ, Streptozotocin; HbA1c, glycated hemoglobin; M/D, Metestrus/Diestrus. Data information: Data depicted here is in biological replicates. In panels (C–I), data is presented as mean ± SEM. In panel (C), data was analyzed using 2-way ANOVA. In panels (D), (F) and (I), data was analyzed using two-way ANOVA with Bonferroni's post hoc test. In panels (E), (G) and (H), data was analyzed using Student's t-test. $*P < 0.05$, $***P < 0.001$; $****P < 0.0001$ ($n = 5$ for control mice, $n = 10$ for STZ mice for panels (C–I)). Source data are available online for this figure.

complications is the level of hyperglycemia (Giacco and Brownlee, 2010), the variable degrees of hyperglycemia in diabetes should also be considered when studying the complications (Service, 2013). Consequently, there is an urgent need to investigate the potential pathogenic effects of the glycemic level achieved in appropriately managed T1D, which has seldomly been done to this date.

We therefore optimized the STZ treatment protocol which resulted in a mouse model that is closer to the glucose dynamics and glycemic control in appropriately managed T1D patients. We used a low-dose STZ administered intraperitoneally over 5 consecutive days. This method better mimics T1D pathogenesis and beta-cell morphological changes compared to the single, high-dose STZ protocol used by many others, whilst also decreasing risk of off-target toxicity-driven collateral tissue damage (Furman, 2021). As female mice are more resistant to STZ-induced diabetes (Saadane et al, 2020), we decided to apply a dosage of 50 mg/kg which we previously have shown to cause severe diabetes in male mice (Zheng et al, 2022) (Fig. 1B). Interestingly, STZ-induced female mice kept a slightly elevated and stable glucose level over the whole experimental period, representing a clinically relevant model for study of the glycemic levels in appropriately managed T1D in female subjects (Fig. 1C). These mice were maintained for up to 10 weeks to mimic the chronic exposure to mild hyperglycemia in patients with juvenile onset of T1D and exclude commonly known potential confounding effects related to STZ administration, such as direct toxicity or STZ-associated systemic inflammation (Fig. 1C). The modest differences in fasting blood between the STZ-treated mice and the control mice were however clearly exposed after oral glucose tolerance test (OGTT) together with an

inappropriate response of the insulin secretion (Fig. 1D–F). The slightly elevated HbA1c in the STZ mice confirmed the chronic exposure of the tissues to mild hyperglycemia (Fig. 1G). Our optimized model hence resembles key clinical features of appropriately managed T1D patients but still with a clear metabolic imbalance compared with control mice.

STZ-induced severe diabetes can lead to detrimental effects on the overall and reproductive health of the maternal mice, resulting in heavy weight loss (Nørgaard et al, 2018) and ovulatory dysregulation shown as disrupted estrous cyclicity (Ryu et al, 2021). We therefore sought to confirm if our mouse model displayed these traits that could potentially confound our following analyses. Reassuringly, our STZ-induced mice showed no difference in either weight (Fig. 1H) or estrous cyclicity (Fig. 1I) at the end of the experiment, demonstrating the suitability of our model to study appropriately managed glycemic levels in subjects with T1D. Importantly, STZ administration can also lead to off-target toxicity, particularly in the kidney due to the expression of GLUT2 (Lenzen, 2008). As we aimed to investigate the long-term effects of glycemic levels corresponding to appropriately managed T1D on the germline and uterine environment, absence of off-target STZ toxicity is crucial. Hence, we analyzed blood from the pregnant mice at sacrifice to examine their overall kidney function. Control and STZ mice had similar levels of hemoglobin as well as several important electrolytes (K+, Na+, Ca2+, Cl-), indicating that our mouse model did not suffer from STZ-related nephrotoxicity with kidney dysfunction (Fig. EV1). Moreover, STZ has a short half-life of 15 min (Wu and Yan, 2015). As we kept the mice for over 10 weeks prior to downstream experimental procedures benefiting

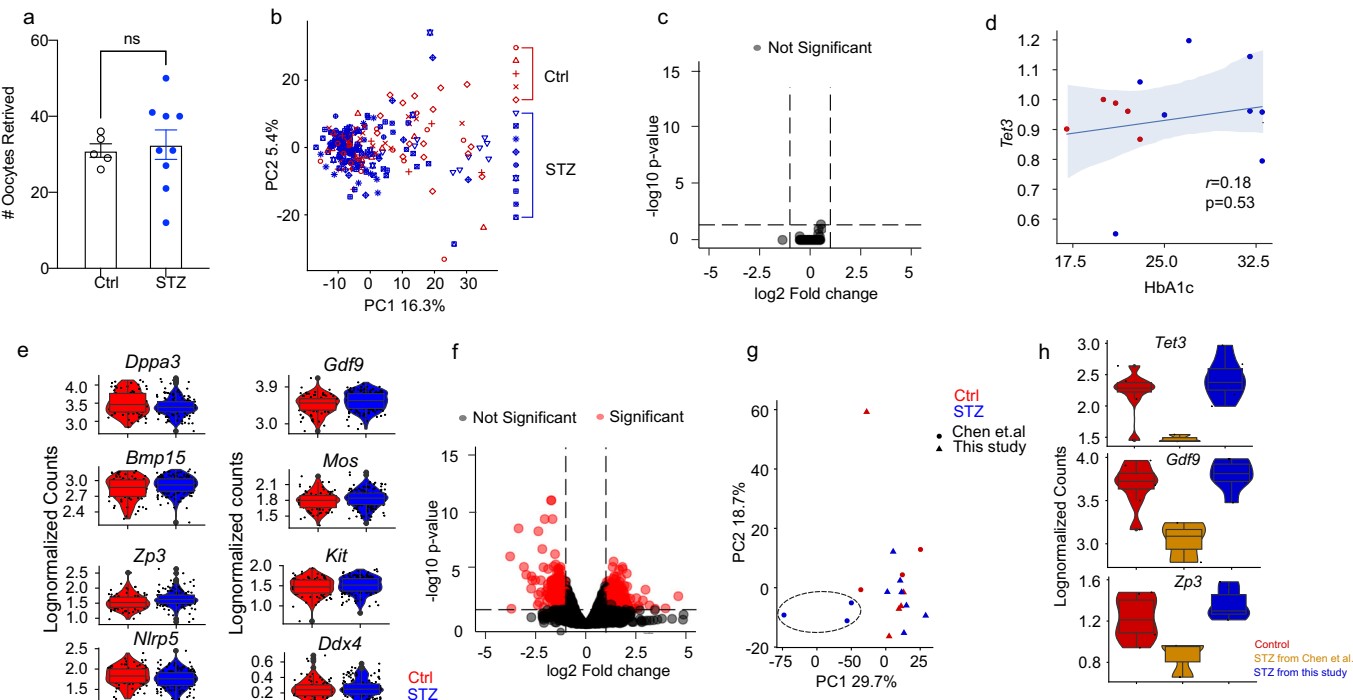

**Figure 2. Mouse oocyte analyses.**

(A) Number of oocytes retrieved after superovulation from Control ($n = 5$) and STZ ($n = 9$) mice. (B) Principal component analysis (PCA) plot of control and STZ oocytes analyzed, also depicting the mouse that the oocyte was obtained from. (C) Volcano plot displaying no differentially expressed genes between Control and STZ oocytes (5 control animals, $n = 81$ oocytes; 9 STZ animals, $n = 149$ oocytes). (D) Correlation between *Tet3* levels and HbA1c levels in oocytes obtained from Control and STZ mice. (E) Violin plots depicting lognormalized readcounts of oocyte-specific genes. (5 control animals, $n = 81$ oocytes; 9 STZ animals, $n = 149$ oocytes). (F) Re-analysis of public transcriptomic data from Chen et al comparing oocytes from poorly managed diabetes and controls ($n = 3$ control minibulk samples, $n = 3$ STZ minibulk samples). (G) PCA plot after integration of data from this study and data from Chen et al, comparing oocytes from poorly managed and appropriately managed diabetes with controls. (H) Violin plots depicting genes downregulated in severe STZ model and retained in mild STZ model ($n = 8$ control samples, $n = 3$ severe STZ samples, $n = 7$ mild STZ samples). Data information: Data depicted here is in biological replicates. In panel (A), data is depicted as mean ± SEM and was analyzed using Student's t-test. In panel (C), data was analyzed using MAST. In panel (F), data was analyzed using Wald test in DESeq2. In panel (D), data was analyzed using a linear regression model. For boxplots in panels e and h, the lower and upper hinges of the box depict the 1st and 3rd quartiles, and the middle line depicts the median. The upper and lower whiskers extend to the largest or smallest value no further than 1.5 times the interquartile range (1.5*IQR) from the hinge. Data beyond the end of the whiskers are plotted as a larger dot, which are then the minimum and/or maximum values depicted. If no large dots are present, the whiskers extend to the minimum and/or maximum values. Source data are available online for this figure.

from their overall improved health, any potential short-term toxic effects would be mitigated. Therefore, our optimized model is valuable for studying the maternal effects of glycemic levels achieved in appropriately managed T1D, which is currently less explored.

Next, we continued to investigate the effects of appropriately managed glycemia on the oocyte transcriptome. As stated previously, we and others have revealed that epigenetic inheritance by maternal disease can be associated with dysregulations of the oocyte transcriptome (Risal et al, 2019; Chen et al, 2022). Furthermore, we wanted to gain further understanding of potential alteration of chromatin accessibility in the oocytes as their maturation is accompanied by gradual chromosomal condensation. Hence, we applied Smart3-ATAC, a method we developed to simultaneously profile the single-cell transcriptome and chromatin accessibility with highest sensitivity (Cheng et al, 2021; Lentini et al, 2022). In total, we collected 272 oocytes from 14 female mice (5 controls, 9 STZ-induced mice). The number of oocytes retrieved from the different groups was similar, different from the severely diabetic STZ model where oocyte retrieval is significantly decreased

(Chen et al, 2022; Lee et al, 2019), further validating the normal reproductive health and ovarian function of our model (Fig. 2A).

After quality control (QC) and filtering, we obtained 230 cells for downstream transcriptomic analysis (Fig. EV2A–D). Interestingly, initial clustering analysis performed using principal component analysis (PCA) failed to display any differences between oocytes derived from the different groups (Fig. 2B). Differential expression analysis further confirmed the lack of differentially expressed genes (DEGs) between the groups, indicating that obtaining glycemic levels reflecting appropriate management in T1D is sufficient to protect the oocyte transcriptome from hyperglycemia-induced alterations (Fig. 2C). When looking more closely into the correlation between different levels of long-term glycemic control and the oocyte transcriptome specifically on *Tet3*, we found no correlation between *Tet3* expression with the HbA1c level of the mouse from which the oocytes were obtained, suggesting that appropriate glycemic levels is sufficient to maintain *Tet3* expression (Fig. 2D). We also confirmed that all MII oocytes expressed normal levels of oocyte-specific genes, such as developmental pluripotency associated 3 *(Dppa3)*, growth differentiation

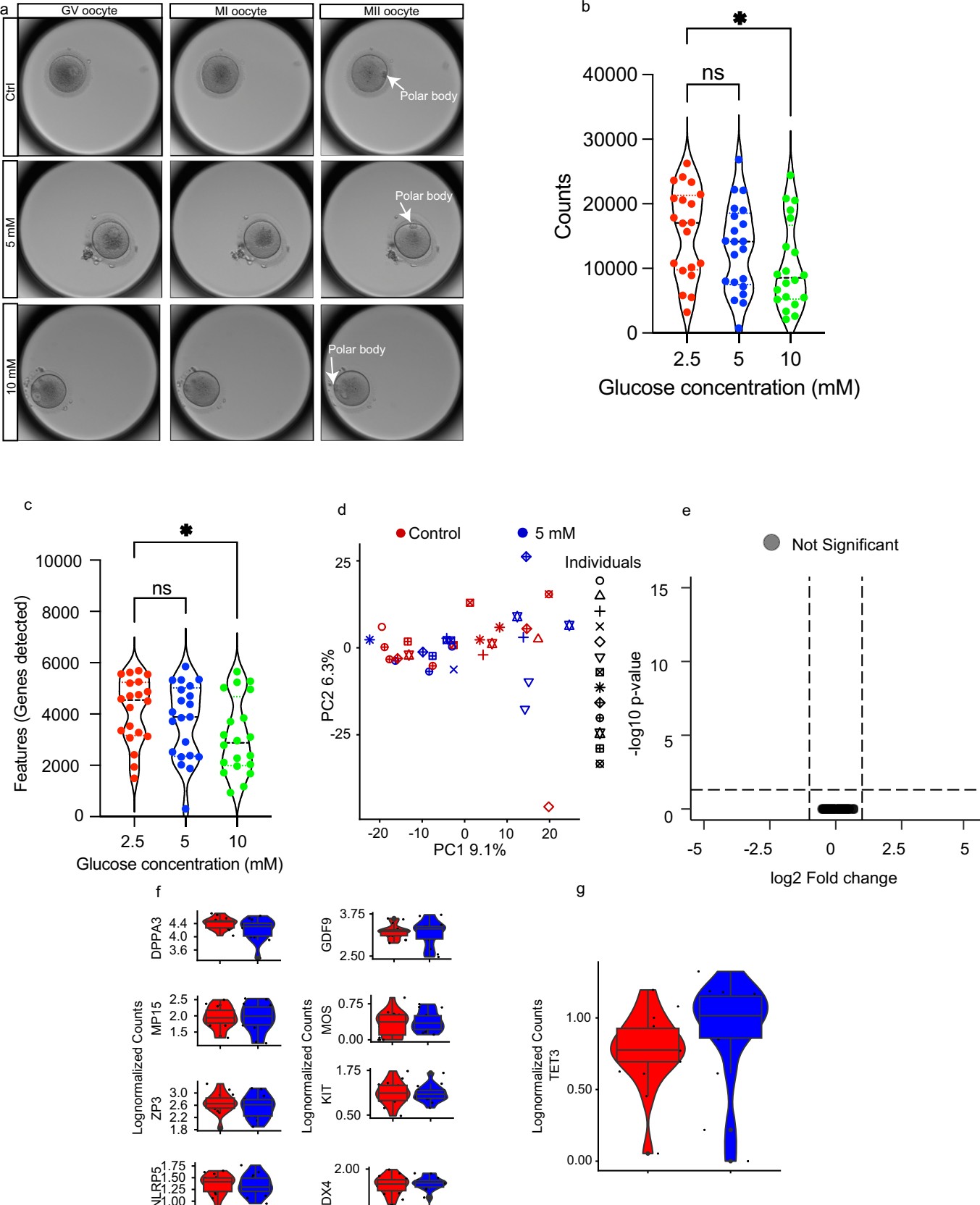

◀ **Figure 3. Human oocyte analyses.**

(A) Representative images for in vitro maturation of oocyte in control (2.5 mM), 5 mM, and 10 mM conditions. Images show the same oocyte in GV (germinal vesicle), MI (Metaphase I), and MII (Metaphase II) stage during maturation. (B) Number of counts (UMIs) for every cell for MII oocytes ($n = 20$ for 2.5 mM, $n = 21$ for 5 mM, $n = 20$ for 10 mM) (C) number of genes/features detected per cell for MII oocytes. (D) PCA plot of oocytes cultured in control (2.5 mM) and mildly diabetic (5 mM) conditions ($n = 20$ for 2.5 mM, $n = 21$ for 5 mM, $n = 20$ for 10 mM). (E) Volcano plot displaying no differentially expressed genes between oocytes undergoing IVM in control and mildly diabetic conditions. (F) Violin plots depicting oocyte-specific genes. ($n = 17$ control oocytes, $n = 16$ 5 mM oocytes). (G) Violin plot depicting *TET3* levels in human oocytes ($n = 17$ control oocytes, $n = 16$ 5 mM oocytes). UMI, Unique Molecular Identifier. Data information: Data depicted here is in biological replicates. In panels (B) and (C), data was analyzed using one-way ANOVA with Bonferroni's post hoc test. In panel e, data was analyzed using MAST. For boxplots in panels (F) and (G), the lower and upper hinges of the box depict the 1st and 3rd quartiles, and the middle line depicts the median. The upper and lower whiskers extend to the largest or smallest value no further than 1.5 times the interquartile range (1.5*IQR) from the hinge. Data beyond the end of the whiskers are plotted as a larger dot, which are then the minimum and/or maximum values depicted. If no large dots are present, the whiskers extend to the minimum and/or maximum values. *$P < 0.05$. Source data are available online for this figure.

factor 9 *(Gdf9)*, bone morphogenic protein 15 *(Bmp15)*, MOS proto-oncogene serine/threonine protein kinase *(Mos)*, zona pellucida glycoprotein 3 *(Zp3)*, KIT proto-oncogene receptor tyrosine kinase *(Kit)*, NLR family, pyrin domain containing 5 *(Nlrp5)* and DEAD box polypeptide 4 *(Ddx4)* (Fig. 2E). For the chromatin accessibility of these oocytes, we obtained 170 cells after QC and filtering (Fig. EV3A–C). The chromatin accessibility was in general low, with low TSS enrichment scores for both control and STZ-derived oocytes, confirming the condensed chromatin properties in MII oocytes (Fig. EV3B). Initial clustering analysis performed by uniform manifold approximation and projection (UMAP) again failed to identify any different clusters among oocytes of the different groups, suggesting similarity in chromatin accessibility (Fig. EV3D). In perfect agreement, there were no differentially accessible peaks found between control and STZ-derived oocytes, further confirming that appropriate glycemic control could be sufficient to prevent hyperglycemia-driven oocyte changes (Fig. EV3E). Altogether, our results provide the first evidence that maintaining glycemic levels corresponding to appropriate glycemic control is effective in safeguarding the oocytes against the detrimental effects caused by hyperglycemia, emphasizing the critical significance of achieving proper pregestational glycemic control.

We next revisited the published transcriptomic datasets from Chen et al (Chen et al, 2022), who identified the role of maternal inheritance of glucose intolerance via oocyte TET3 insufficiency by a mouse model of severe diabetes and hyperglycemia. We compared the effects of appropriately- and poorly managed glycemia on the oocyte transcriptome (Chen et al, 2022). We found that severe hyperglycemia induced by the high-dose STZ resulted in abundant transcriptomic alterations with 717 DEGs, displaying substantial disparity compared to our results and further emphasizing the pronounced differences between the impacts of appropriate and poor glycemic control on oocytes (Fig. 2C,F). To further compare global gene expression with oocytes from Chen et al, dataset, we also performed pseudobulk expression profiling through summarizing the gene counts in our oocytes before data integration for the clustering analysis. In line with our previous results, a clear separation was identified for oocytes exposed to poorly managed glycemia by PCA, whilst oocytes from our model were intermingled with both our and their control samples, implying similarity in transcriptomic profile among these oocytes (Fig. 2G). We also observed that apart from *Tet3*, both *Gdf9* and *Zp3* were downregulated in oocytes from Chen et al, indicating overall poor quality as oocyte-specific genes were altered (Fig. 2H).

Importantly, we also observed that the expression level of these genes was normalized in oocytes obtained from females with appropriate glycemic levels (Fig. 2H).

To further validate our findings in human conditions, we sought to investigate if appropriate management of glycemic levels could preserve the transcriptomic signatures in human oocytes. Importantly, thanks to advancements in diabetic care, T1D patients now experience lower rates of fertility issues (Jonasson et al, 2007). Consequently, the retrieval of oocytes from T1D patients for research purposes is ethically inappropriate. We therefore chose to study the effects of mild hyperglycemia reflecting appropriate glycemic control on the oocyte during the maturation process using immature oocytes obtained from non-diabetic patients undergoing clinical IVF procedure followed by in vitro maturation (IVM) cultured with different glucose concentrations. We cultured the oocytes using 2.5 mM glucose as a control environment, and 5 mM as a diabetic environment based on previous follicular fluid measurements displaying these concentrations to be average for individuals without and with diabetes, respectively (Chen et al, 2022). Importantly, higher glucose concentrations (10 and 15 mM) during IVM have previously been demonstrated to affect the oocyte transcriptome and *TET3* expression, but IVM in 5 mM glucose has never been investigated (Chen et al, 2022). As comparison, we also matured oocytes in 10 mM glucose to check the previously identified effect on the oocyte transcriptome during IVM at this glucose level. We found that these oocytes matured in all conditions, without any obvious macroscopic differences (Fig. 3A). After maturation to MII oocytes, we proceeded to analyze oocytes at a single-cell level using Smart-seq3 to profile their transcriptome.

In total, we collected 67 oocytes from 23 patients, which were randomly distributed into the three conditions. Next, we analyzed the transcriptomic profile of each oocyte among the different groups, examining the total amount of counts sequenced per cell and total amount of genes detected. There were no significant differences between oocytes cultured in 2.5 mM and 5 mM for these parameters, suggesting that oocytes cultured in 2.5 mM and 5 mM glucose maintained similar transcriptomic integrity (Fig. 3B,C). However, oocytes cultured with 10 mM glucose were of significantly worse quality transcriptome-wise and signs of degradation, with both less genes and read counts per cell (Fig. 3B,C). Hence, IVM culture at very elevated glycemic level of 10 mM glucose could cause RNA degradation and decrease the transcriptomic quality of the oocytes, whilst culture at a slighter increase of glycemic levels of 5 mM did not have the same effect.

Even if IVM at 5 mM did not affect the overall transcriptome like IVM at 10 mM did, it still might have affected the expression of specific genes, altering the transcriptome in a different way without affecting the overall quality. Therefore, we further investigated the differential gene expression of oocytes cultured in 2.5 mM and 5 mM conditions to identify potential differences. After quality control, 33 oocytes (17 in the 2.5 mM group and 16 in the 5 mM group) remained for downstream transcriptomic analysis (Fig. EV4A–D). Similar to the mouse oocytes in our model, PCA failed to separate oocytes cultured in 2.5 mM and 5 mM, suggesting that a slight increase of glucose in the oocyte environment does not alter their transcriptome (Fig. 3D). In line with this, we did not find any significantly altered gene expression between human oocytes cultured in 2.5 mM and 5 mM glucose during maturation (Fig. 3E). Oocyte-specific genes as mentioned above were expressed at similar levels in oocytes matured in both 2.5 mM and 5 mM glucose (Fig. 3F). Moreover, *TET3* expression was not significantly altered in oocytes matured in 2.5 mM and 5 mM glucose for both adjusted *p*-value and raw *p*-value ($p > 0.05$ for both) (Fig. 3G). Taken together, these results aligned with our findings from the mouse model, further validating that oocyte transcriptome integrity was maintained when exposed to a mild increase in glycemic levels reflecting appropriate glycemic management throughout their growth and maturation environments, once again emphasizing the importance of pregestational glycemic care for women with T1D.

As offspring of women with T1D still suffer a higher risk of adverse health outcomes such as metabolic dysfunction unrelated to maternal glycemic management measured by HbA1c during pregnancy (Vlachová et al, 2015), and since we found the germline to be largely unaffected by glycemic levels and dynamics corresponding to appropriately managed T1D, we hypothesized that the uterine environment might be dysregulated, resulting in altered fetal development and increased disease susceptibility. Therefore, we proceeded to investigate fetal growth and placentas in our model of appropriately managed glycemia to gain further insights. We successfully mated STZ and control females with healthy male mice and collected the fetuses and placentas at embryonic day 18.5 (E18.5), when fetuses are near term and the placenta is fully mature (Elmore et al, 2022). Maternal STZ dams had a slight elevation in HbA1c levels at E18.5 (Fig. 4A) comparable to the values before mating (Fig. 1G), indicating that the appropriately managed glycemic state was kept during pregnancy. Intriguingly, the fetuses of STZ dams were significantly smaller, demonstrated by a reduced crown-rump length and weight compared to controls (Fig. 4B–D). Moreover, the ratio of fetal resorption was nearly twice more in STZ dams compared to controls, further confirming a disturbed uterine environment in STZ pregnancies (Fig. 4E,F). Interestingly, although the placental gross morphology and weight was similar between the two groups (Fig. 4G,H), the placental efficiency, defined as the ratio between the weight of the fetus and the placenta, was lower in the STZ group (Fig. 4I) implicating that placental dysfunction could be underlying the fetal growth deviation. These findings suggest that maintaining glycemic levels corresponding to appropriate management in T1D is not sufficient to protect the uterine environment, instead causing placental dysfunction and altering fetal growth.

We next sought to understand the potential changes in the placenta that could have led to its dysfunction. The labyrinth zone of the mouse placenta acts as the site of nutrient exchange, and any defects in this zone might therefore underlie fetal growth deviations (Woods et al, 2018). We therefore performed hematoxylin and erythrosine staining to analyze the morphology and size of the labyrinth zone in control and STZ placentas. Our analysis showed no significant difference in labyrinth size between the two groups, with the overall structure also being maintained (Fig. 5A). We therefore next sought to identify potential molecular changes underlying placental dysfunction using bulk RNA sequencing. The principal component analysis (PCA) based on the global gene expression of each placenta showed that placentas mostly clustered according to their groups suggesting consistent effects of maternal T1D (Fig. 5B). In total, 96 DEGs were identified using DEseq2 with most of them (84/96) being upregulated (Fig. 5B; Table EV1). One of the top upregulated DEGs was *Vegfd*, suggesting upregulated angiogenesis in STZ placentas (Fig. 5C,D). In addition, we found an upregulation of endothelial-cell-specific genes such as *Pecam1* (CD31) (DeLisser et al, 1997) and *Srgn* (Kolset and Tveit, 2008), further implicating an increased angiogenesis and a higher population of vessel endothelial cells in STZ placentas (Fig. 5C,D). Another highly upregulated gene was *Lox*, which is a direct target of the HIF-1a, a key regulator of oxygen homeostasis in response to hypoxia (Erler and Giaccia, 2006) (Fig. 5C,D). Interestingly, diabetes has previously been associated with increased hypoxic pressure (Catrina and Zheng, 2021), and hypoxia is also a strong promoter of angiogenesis (de Heer et al, 2020) and also known to cause fetal growth impairment (Colson et al, 2021). Hence, the STZ placentas might have been exposed to chronic hypoxia, which would further affect the fetal growth observed in our model (Fig. 4B–D).

We also performed gene set enrichment analysis (GSEA) to further dissect our transcriptomic data. Reassuringly, we found that the hallmark terms of angiogenesis and hypoxia were among those terms that were significantly enriched, further implicating increased hypoxic pressure as potential pathogenic factors underlying the placental dysfunction and fetal growth restriction in STZ dams (Fig. 5E). To validate the gene expression changes of angiogenesis in the placenta, we performed immunofluorescent staining of CD31, a widely used marker for endothelial cells, and quantified the amount of CD31[+] cells in the placental junctional zone, which normally is less vascularized compared to the labyrinth zone (Woods et al, 2018). In line with the findings of increased angiogenesis observed in the RNA sequencing analysis, we observed a higher proportion of CD31[+] cells in the placentas of the STZ group compared to controls (Fig. 6A). We then further analyzed whether the upregulation of angiogenesis might occur as a consequence of chronic hypoxia as we hypothesized. To do so, we injected pregnant females with pimonidazole intraperitoneally 2 h prior to sacrifice. Pimonidazole forms covalent bonds with cellular macromolecules at oxygen levels below 10 mmHg to indicate poorly oxygenated regions (Labiano et al, 2022). By pimonidazole immunofluorescence staining, we found that the average intensity of pimonidazole signal was elevated in the junctional zone of the placentas from the STZ group compared to controls, validating an increase of hypoxic environment in placentas associated with appropriately managed T1D (Fig. 6B). Collectively, these findings suggested that glycemic levels reflecting appropriately managed maternal T1D still impairs the uterine environment as a result of placental hypoxia and dysfunction,

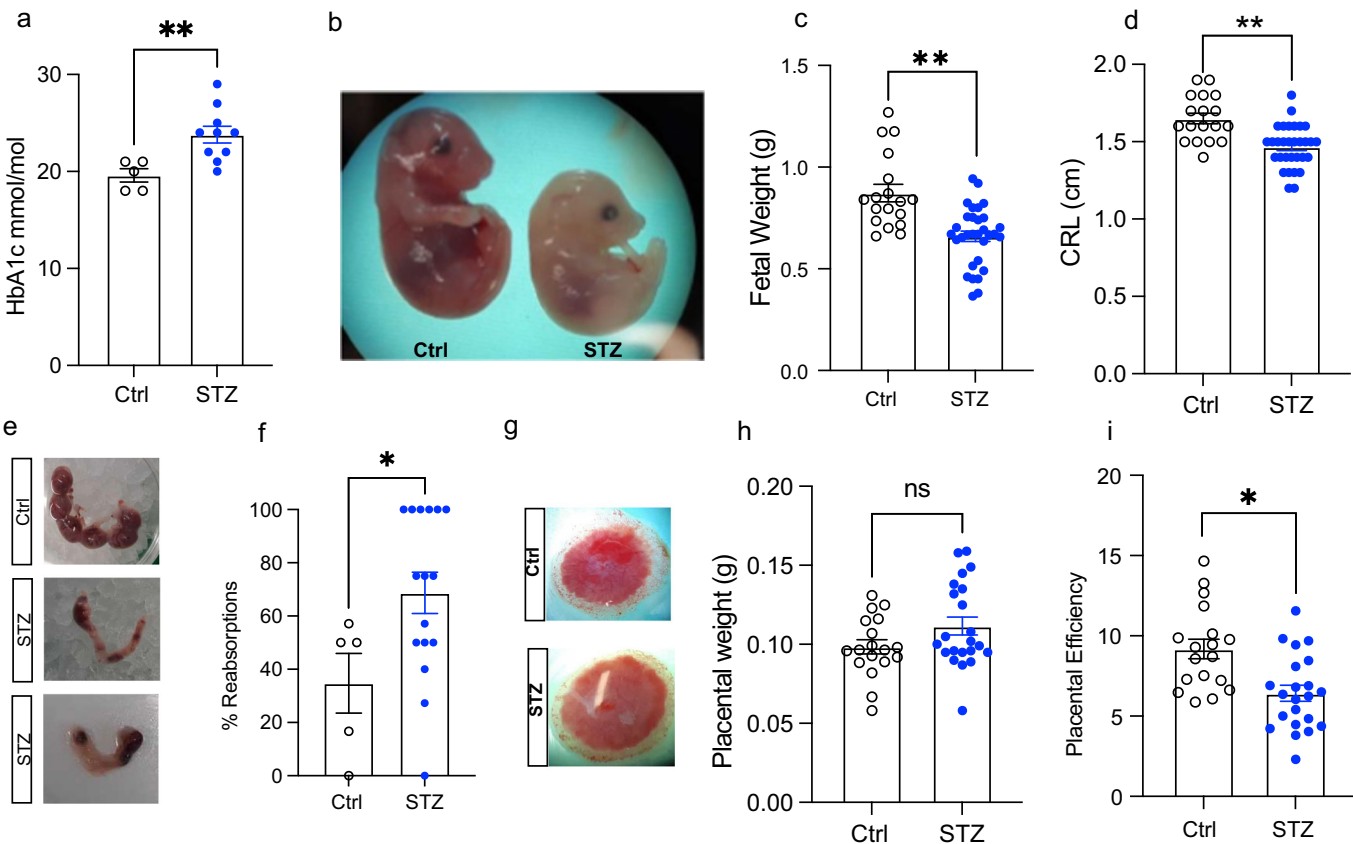

**Figure 4.  Mouse fetal development study.**

(**A**) HbA1c levels at end of experiment ($n = 5$ control mice, $n = 10$ STZ mice). (**B**) Representative image for Control (left) and STZ (right) embryo. (**C**) Embryo weight of embryos derived from Control ($n = 18$) and STZ ($n = 31$) mice. (**D**) Crown-rump length (CRL) of control ($n = 18$) and STZ ($n = 31$) embryos at E18.5. (**E**) Representative images of uteruses at E18.5. Uterus with no reabsorption (top, Control) and with several both early and late reabsorptions (middle and bottom, STZ). (**F**) Percentage of embryos reabsorbed at E18.5 ($n = 5$ control and $n = 16$ STZ). (**G**) Representative images for placentas at E18.5. (**H**) Placental weight from control ($n = 18$) and STZ ($n = 22$) fetuses at E18.5. (**I**) Placental efficiency of control ($n = 18$) and STZ ($n = 22$) placentas linked to embryos depicted in (**B**). STZ, Streptozotocin; HbA1c, glycated hemoglobin; CRL, Crown-rump length. Data information: Data depicted is biological replicates. In panels (**A, C, D, F, H, I**), data is depicted as mean ± SEM. In panels (**A**) and (**F**), data was analyzed using Student's t-test. In panels (**C**), (**D**), (**H**) and (**I**) data were analyzed using analysis of covariance (ANCOVA) to correct for litter size. *$P < 0.05$, **$P < 0.01$. Source data are available online for this figure.

consequently affecting the growth and development of the fetus and potentially priming the offspring for metabolic disease later in life.

## Discussion

Hyperglycemia is consensually considered to play a key role in the development of diabetes complications, and glycemic control in T1D patients has steadily improved over time. This highlights the urgent need to investigate how glycemic levels reflecting improved management of T1D affect the germline and uterine environment in T1D. A noteworthy and worrisome discovery has previously uncovered significant changes in gene expression in oocytes affected by severe T1D, particularly highlighting the implications of *TET3* insufficiency to be associated with glucose intolerance in offspring (Chen et al, 2022). The implication of these findings thereby raises a profound concern regarding the potential long-term impact of maternal hyperglycemia on health across generations. However, the translational relevance of these findings is

questionable due to the non-clinically relevant levels of extreme hyperglycemia as well as the presence of several confounding factors within the utilized model such as overall weight loss, infertility, and ovulatory dysfunction (Chen et al, 2022; Lee et al, 2019; Ryu et al, 2021). The glycemic control in subjects with T1D of reproductive age has increasingly improved over the last years, likely contributing to enhanced general health including reproductive function and pregnancy outcomes (Fig. 1A). Hence, it is more clinically relevant to investigate the effects of glycemic levels corresponding to appropriately managed T1D on the germline and uterine environment, a condition which is understudied to this date. Our results demonstrated that a slight, but significant, elevation of glucose levels during follicle growth and oocyte maturation, similar to levels often seen in appropriate T1D management, preserves the transcriptomic integrity of oocytes and is not sufficient to cause hyperglycemia-induced damage as seen previously. Thus, we highlight the significance of proper pregestational control to prevent oocyte-driven epigenetic inheritance of disease in maternal T1D.

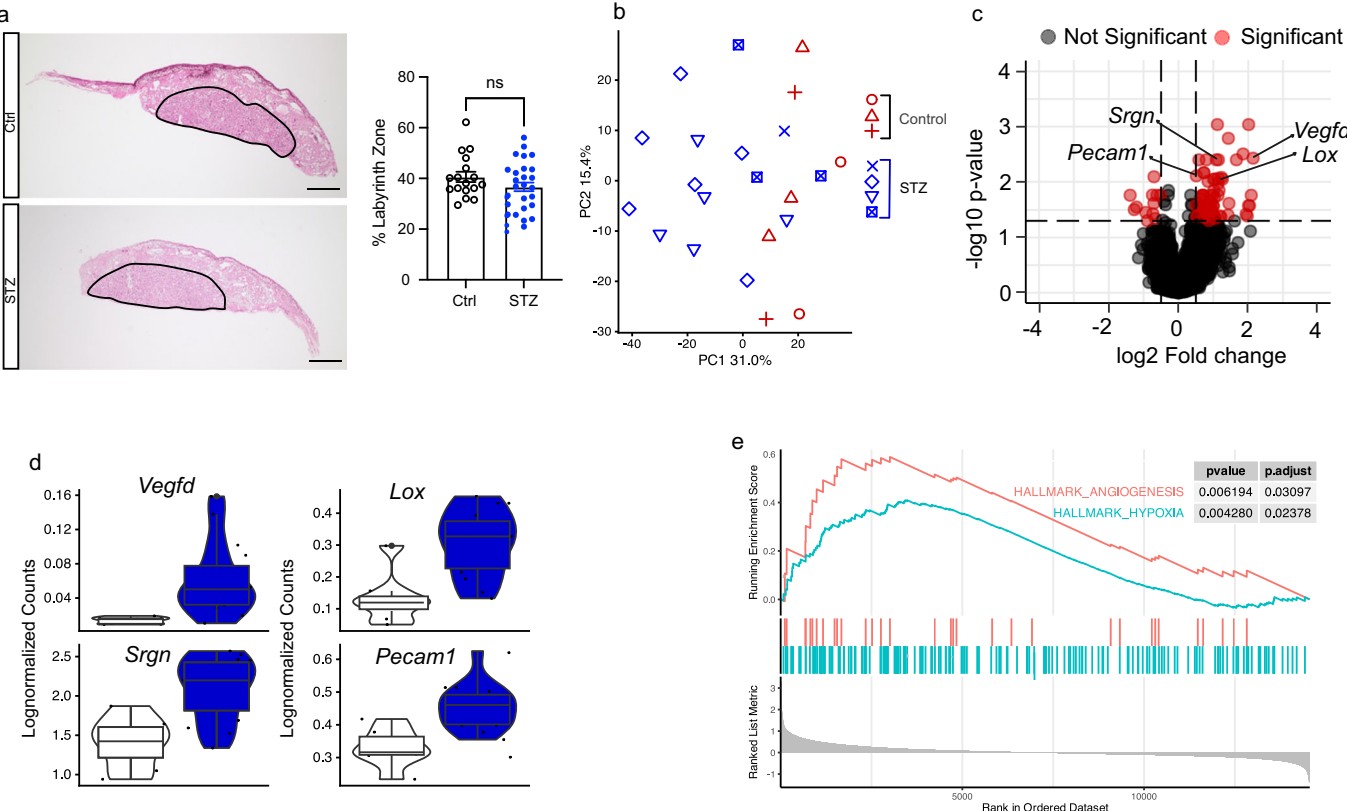

**Figure 5. Histological and transcriptomic analysis of placentas.**

(A) Hematoxylin and erythrosine staining with representative images and quantification analyzing labyrinth zone size in control ($n = 17$) and STZ ($n = 30$) placenta. Scale bar: 800 μm. (B) Principal component analysis (PCA) plot of control ($n = 8$) and STZ ($n = 14$) placenta RNAseq, symbols depicting which placentas belong to the same maternal mouse. (C) Volcano plot depicting results of differential expression analysis between control ($n = 8$) and STZ ($n = 14$) placentas. (D) Violin plots depicting individual gene expression of genes highlighted in (C)) in control ($n = 8$) and STZ ($n = 14$) placentas. (E) Enrichment plots of the hallmark Angiogenesis and Hypoxia gene sets generated by GSEA comparing control ($n = 8$) and STZ ($n = 14$) placentas. STZ, Streptozotocin; PCA, principal component analysis; GSEA, gene set enrichment analysis. Data information: Data depicted is biological replicates. For panel (A), data is depicted as mean ± SEM and was analyzed using Student's t-test. For panel (C), data was analyzed using Wald test in DESeq2. For boxplots in panel (D), the lower and upper hinges of the box depict the 1st and 3rd quartiles, and the middle line depicts the median. The upper and lower whiskers extend to the largest or smallest value no further than 1.5 times the interquartile range (1.5*IQR) from the hinge. Data beyond the end of the whiskers are plotted as a larger dot, which are then the minimum and/or maximum values depicted. If no large dots are present, the whiskers extend to the minimum and/or maximum values. *$P < 0.05$. Source data are available online for this figure.

Our comparison with previous datasets further demonstrated the distinct outcomes between poor and appropriate glycemic control when it comes to oocyte transcriptome. In addition to differences in blood glucose concentrations, our refined T1D mouse model offers the advantage of being free from the other confounding factors observed in previously utilized diabetic mouse models in this field of study. Most notably, STZ-induced mice are frequently employed for short-term studies due to the development of severe illness with time, which potentially introduces concerning factors associated with the toxicity of STZ administration. Indeed, many studies use STZ-induced mice within a few weeks after injection, and one example of this is the study by Chen et al, which maintained their mice for only 30 days before collecting MII oocytes for IVF studies (Chen et al, 2022). Apart from toxic and off-target effects on other organs such as the kidney as mentioned previously, STZ-induced diabetes is also associated with increased inflammation after STZ injection, both locally in the pancreas (Rossini et al, 1977; Weide and Lacy, 1991) and more systematically with increased levels of proinflammatory cytokines (Bathina et al,

2017; Arokoyo et al, 2018; Niu et al, 2016) and dysregulated immune cell responses (Lee et al, 2016). It is important to consider that hyperglycemia is pro-inflammatory in its nature, hence the systematic inflammation in STZ-injected mice could partially be due to hyperglycemia-related inflammation (Giacco and Brownlee, 2010). However, STZ has previously been demonstrated to have direct effects on lymphocytes through which STZ can alter the immune system homeostasis acutely post-injection through mechanisms independent of hyperglycemia (Muller et al, 2011). These acute effects on the immune system homeostasis could subsequently contribute to the inflammatory response. STZ injection can also cause a strong inflammatory response through its off-target toxicity on organs other than the pancreas, also independent of hyperglycemia (Muller et al, 2011). Therefore, a longer maintenance period prior to experimental procedures could greatly mitigate these potential confounding factors related to STZ administration, allowing for studying the sole effect of hyperglycemia. Importantly, our STZ mice can be maintained in a healthy condition for a minimum of 10 weeks before the following

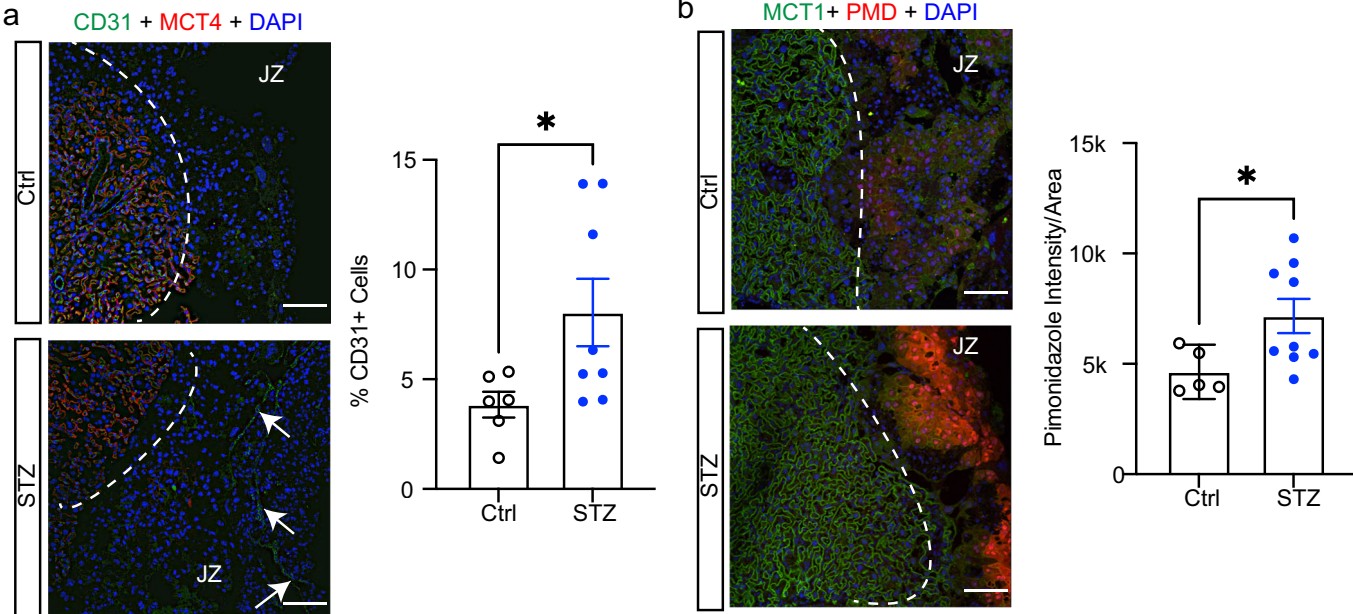

**Figure 6. Confirmation of hypoxia and angiogenesis using immunohistochemistry in placentas.**

(A) Fluorescent immunohistochemistry with representative images and quantification analyzing control ($n = 6$) and STZ ($n = 8$) placenta stained with CD31 (green, endothelial cell marker), MCT4 (red, placenta labyrinth zone marker) and DAPI (blue, nuclear staining). White arrows point towards CD31-positive staining in placenta. Scale bar: 100 µm. (B) Fluorescent immunohistochemistry with representative images and quantification analyzing control ($n = 5$) and STZ ($n = 9$) placenta stained with Pimonidazole (red, hypoxia marker), MCT1 (green, placenta labyrinth zone marker) and DAPI (blue, nuclear staining). Scale bar: 100 µm. STZ, Streptozotocin; MCT4, Monocarboxylate transporter 4; MCT1, Monocarboxylate transporter 1. Data information: Data depicted is biological replicates. For panels (A) and (B), data is depicted as mean ± SEM and was analyzed using Student's t-test. *$P < 0.05$. Source data are available online for this figure.

experiments. This extended duration not only mirrors the interval between juvenile onset of T1D to conceiving time but also offers a significant advantage in mitigating non-specific toxic effects of STZ administration for downstream analyses. Notably, our mice display normal levels of hemoglobin (Fig. EV1). As chronic inflammation is often associated with anemia, a similar phenotype in our mice would be expected if they displayed chronic inflammation related to STZ injections. Therefore, the results obtained in our study are most likely as a result of the mild hyperglycemia observed.

The different methods of STZ administration in our study compared to that used by Chen et al (i.e., multiple injections with a lower concentration versus one injection with high concentration) might differ in off-target toxic effects, as discussed previously. Moreover, the severe hyperglycemia itself and the potential increased systemic inflammation in STZ-induced severely hyperglycemic mice in Chen et al's paper might compromise the oocyte maturation and estrus cyclicity of the mice, leading to overall poorer oocyte transcriptome quality and extensive alteration of gene expression as shown in their study (Chen et al, 2022). We have previously discussed the potential increased systemic inflammation in STZ-induced severely hyperglycemic mice, which also could affect reproductive function which was not characterized in their study as we have done here. Notably, the mice used by Chen et al indeed displayed increased levels of serum testosterone (Chen et al, 2022). As many studies including ours have demonstrated transcriptome-altering effects in oocytes in mouse models of polycystic ovary syndrome characterized by hyperandrogenism (Risal et al, 2019), this elevation of serum testosterone could also

contribute to alterations of the oocyte transcriptome. It is however important to note that many of these other confounding factors could develop as a result of extreme hyperglycemia, which further highlights how lowering of the glycemic levels is crucial to protect the oocyte transcriptome.

Notably, our results showed that despite appropriate glycemic management, maternal T1D can still affect the uterine environment, likely because the placenta is a richly infused organ with blood in both humans and rodents as well as being metabolically very active, consuming 40–60% of the oxygen and glucose delivered to the uterus at the end of the gestation (Maltepe and Fisher, 2015; Wang and Zhao, 2010; Buelke-Sam et al, 1982). Therefore, even small changes in nutrient availability and uptake can have greater effects when compared to the ovary, which is less perfused, and hence the placenta could be more affected by the slightly elevated glycemic levels, subsequently affecting the supply of nutrients and gases to the fetus. The direct exchange of oxygen and nutrients through the uteroplacental circulation with the maternal blood flow into the intervillous space in the placenta tightly associates the maternal physiological condition with the uterine environment. Our findings pinpoint that the placental hypoxic environment with compensatory upregulation of angiogenesis caused by glycemic levels corresponding to appropriately managed T1D could have a significant impact on fetal growth. Moreover, one of our top upregulated DEGs is *Lox*, which has previously been shown to compromise permeability of endothelial cells in the retina of diabetic mice (Chronopoulos et al, 2010). The ectopic vessel formation that we identified in this study might therefore not be

functional, which could explain why the compensatory upregulation of angiogenesis did not alleviate the placental hypoxia. Though disheartening, this discovery represents a crucial step toward comprehending why offspring of T1D women with appropriate glycemic control may still experience adverse developmental effects (Ludvigsson et al, 2018, 2019; Vlachová et al, 2015). This also pinpoints the need of additional therapy specifically during the gestational period even for women with appropriately managed T1D. Further investigations are now imperative to unravel the long-term effects of glycemic levels corresponding to appropriately managed T1D on the health of adult offspring and explore potential preventive measures to target the dysregulated gene- and signaling pathways in placenta and mitigate such epigenetic inheritance by intervening in the uterine environment.

Notably, as with most other mouse models, an important disparity of our mouse model with appropriately managed T1D in humans is lack of exogenous pharmacological treatment (most often insulin), which is challenging to achieve in mouse models. However, the key feature on the glycemic levels and dynamics in patients with appropriately managed T1D is well mirrored by our mice. Moreover, our IVM culture of human oocytes to model the diabetic follicular environment is also not without limitations when compared to oocytes from patients with T1D, since it encompasses the final maturation stages of oocytes corresponding to transition from GV to MII oocytes. It is noteworthy that this frame time of IVM encompasses relevant processes of genomic reorganization, following a previous long latent stage. Consequently, this brief window could presumably be more susceptible to genomic changes. Furthermore, the possibility to obtain mature oocytes from T1D patients solely for research purposes is both ethically and logistically challenging, and as few patients require assisted fertility treatment of this date, it is also difficult to obtain oocytes in conjunction with the IVF process. We do, however, believe that in vivo data from the mouse model together with human in vitro data still provides convincing evidence to support our conclusions.

Our findings collectively demonstrate that maintaining glycemic levels corresponding to appropriate glycemic control is essential for T1D patients to preserve oocyte transcriptomic profile but not sufficient to prevent alterations in fetal growth and development, instead causing an adverse uterine environment and placental dysfunction. These results underscore the significance of implementing an effective pregestational therapeutic regimen tackling hyperglycemia and serve as a compelling motivating factor for young female patients with T1D to strive for optimal glycemic control. Moreover, these results highlight the need for future therapeutic approaches on the placenta during pregnancy to prevent detrimental effects of the uterine environment when it comes to developmental programming.

# Methods

## Study of glycemic trends in females of reproductive age

Data were extracted from the National Diabetes Register in Sweden (Nationella Diabetesregistret, NDR) using the open-access online tool *Knappen 2.0* (https://www.ndr.nu/Knappen2/?apiURL=https://www.ndr.nu/api/) (Nationella Diabetesregistret). Data for the figures were extracted on the 20th of June 2023. The following parameters were used: Adult care, female, type 1 diabetes, HbA1c < 52 mmol/mol or HbA1c > 70 mmol/mol, and 18–45 years of age. Data were then depicted as the proportion of patients fulfilling the criteria in %, with 95% confidence interval. Ethical approval was not required due to the public nature with open-access of group-level data in *Knappen 2.0*.

## Animals

Nine-week-old female C57BL/6J mice were obtained from Janvier Labs. All mice were maintained under a 12-h light/dark cycle in a temperature-controlled room with ad libitum access to water and food. At 10 weeks of age, diabetes was induced using 5-day consecutive intraperitoneal injections of Streptozotocin (Sigma-Aldrich). Mice were randomly allocated to control or diabetes groups based on baseline glucose measurements and weights. Weighing was done weekly from week 0. Tail vein morning glucose measurement (Freestyle Precision) was done weekly from week 5 after induction to monitor the stability of the model. After 10 weeks of disease exposure, mice were used for mating or oocyte collection.

The rest of the dams were mated with healthy male mice and sacrificed through cervical dislocation under isoflurane anaesthesia on gestational day E18.5. Females were checked for post-copulatory plugs, and a plug on the morning after mating was considered E0.5. For later analyses of placental hypoxia, a pimonidazole solution (Hypoxyprobe-1 Omni Kit, Hypoxyprobe) was administered intraperitoneally at a dosage of 60 mg/kg body weight 120 min prior to tissue harvest as previously described (Zheng et al, 2022). The placentas and embryos were collected and weighed. Placental efficiency was calculated as the ratio of the embryonal weight to the placental weight. One-quarter of each placenta was fixed in 4% paraformaldehyde and embedded in paraffin for histological analyses. The remaining three quarters were snap frozen in liquid nitrogen and stored at −80 °C for RNA-analyses. The crown-rump length, measured from the top of the head to the bottom of the rump, of the embryos was assessed. To determine the number of resorptions, reflecting the number of intrauterine embryonal deaths, visual inspection of the uterus was performed. Analysis of all animal phenotypes was blinded.

## Measurement of estrus cyclicity

For assessment of the reproductive function, the oestrus cyclicity was characterized in STZ and control dams 10 weeks following the final injection of STZ or PBS using the method previously described by Caligioni (Caligioni, 2009). For 10 consecutive days, flushing of the vagina was performed with 15 μl NaCl. The cellular composition of the vaginal secretion was assessed daily using a light microscope (Leica) at 10× magnification. Based on the proportions of nucleated epithelial cells, anucleated cornified cells and leukocytes, the stage of the oestrous cycle was determined. Proestrus was defined by the presence of predominantly nucleated epithelial cells and estrus by anucleated cornified cells. All three cell types were present in metestrus whereas diestrus was characterized by the presence of predominantly leukocytes. To enable comparison between control and STZ mice, the average time spent in each stage of the oestrus cycle was calculated. Metestrus and diestrus were combined for analysis purposes. Analysis of estrus cyclicity was blinded.

## Oral glucose tolerance test

Glucose metabolism was measured by an OGTT after a 5 h fast at 18 weeks of age (8 weeks after induction). D-glucose was administered orally by gavage (2 g/kg) and blood glucose was measured at time 0, 15, 30, 60, and 90 min. 100 µL of blood was collected at 0 and 15 min for insulin measurement.

## Serum insulin measurement

Serum insulin of blood samples obtained from the OGTT was analyzed using an ELISA kit (Crystal Chem) according to the manufacturer's instructions. 5 µL of serum was used for analysis.

## Blood electrolyte and hemoglobin measurement

Levels of electrolytes and hemoglobin were measured in blood samples collected post sacrifice of STZ-induced mice. The measurements were performed on an ABL800 device (Radiometer) according to the manufacturer's instructions.

## Oocyte retrieval in mice

For oocyte retrieval, 20-week-old mice (10 weeks after induction) were superovulated through injection with 100 µL of CARD HyperOva (Cosmo Bio Co., Ltd) followed by 5 IU of human chorionic gonadotropin (hCG; Chorulon Vet 1500 IE, MSD Animal Health Care) 48 h after priming. Cumulus-oocyte complexes were subsequently isolated 16 h after hCG injection from oviduct ampulla. Denuded single MII oocytes were obtained by removing the cumulus mass in M2 Medium (M7167, Merck), containing 0.3 mg/ml hyaluronidase (H3884, Merck) at RT. Single oocytes were then put into lysis buffer in 0.2 mL strips for subsequent library preparation.

## Human hormonal stimulation treatment and oocyte retrieval

Immature oocytes at GV or MI stages were donated by healthy non-diabetic women aged 25–39 years after signed informed consent. The women were undergoing controlled ovarian stimulation (COH) treatments aimed at oocyte cryopreservation or intracytoplasmic sperm injection (ICSI) at the Department of Reproductive Medicine of Karolinska University Hospital. The standard COS treatments and clinical routines have been previously described (Feichtinger et al, 2017). Briefly, either a long-protocol, using a nasal GnRH agonist (Nafarelin 800 µg daily; Synarela, Pfizer) or buserelin 1200 µg daily; Suprecur, Sanofi), or a short protocol using GnRH antagonist (0.25 mg once daily, SC ganirelix, Fyremadel, Ferring) initiated routinely on the fifth day of COS were conducted. The dose of COS (75–400 IU daily of recombinant follicle stimulating hormone (Gonal-F Merck or hMG Menopur, Ferring) was individualized to patient's age, menstrual cycle length, antral follicle count and anti-Mullerian hormone levels. Ovarian follicle growth tracking was performed by scheduled transvaginal ultrasound examinations that lead to the planning timepoint for oocyte maturation triggered by administration of recombinant hCG (250 µg SC Ovitrelle; Merck). Oocyte retrieval was carried out by transvaginal ultrasographically-guided follicular puncture 37 h later.

## Human oocyte denudation and maturity assessment

Cumulus-oocyte complexes (COCs) were retrieved from the follicular fluid under stereomicroscope, collected and incubated for further processing in G-IVF medium at 37 °C, 6% $CO_2$ 95% humidity. Denudation of COCs was carried out after brief exposure to 80 IU/mL hyaluronidase (Vitrolife, Sweden) and followed by mechanical disaggregation of granulosa cells that once removed enabled maturity assessment of oocytes.

## Human in vitro maturation (IVM) of oocytes

Immature oocytes at stages GV or MI after denudation donated for this study were anonymized and thereafter placed into culture medium for in vitro maturation (IVM) for 48 h maximum. Oocytes were randomly allocated to one of three different glucose conditions on G-IVF culture medium: standard 2.5 mM glucose condition as control group, 5 mM glucose supplementation as a mildly diabetic group, and 10 mM glucose condition as a severely diabetic group. Culture of oocytes was monitored under time-lapse imaging incubator (Geri, Genea Biomedx) under 37 °C, 6% $CO_2$, 5%$O_2$ and 95% humidity. Times to germinal vesicle breakdown (GVBD) GV → MI and time to maturation (GV → MII) were individually recorded for GV and time to first polar body (PBI) extrusion starting from the moment they were placed in culture and recorded with time interval of 5 min. Once matured, individual oocytes were exposed to Tyrode's acid for zona pellucida removal washed in PBS 0.2% HSA and loaded into single tubes.

## Smart3-ATAC single cell library preparation and sequencing

Single oocyte multi-omic libraries were prepared using the Smart3-ATAC protocol as previously described (Cheng et al, 2021). Briefly, oocytes were put into 4 µL of lysis buffer before centrifugation at $1900 \times g$ for 5 min at 4 degrees. Thereafter, samples were allowed to rest on ice before vortexing at 1000 rpm for 3 min and centrifugation at $1900 \times g$ for 5 min at 4 degrees. Thereafter, the supernatant was carefully separated from the nucleus and snap-frozen for later mRNA library preparation using Smart-seq3 (Hagemann-Jensen et al, 2020). scATAC libraries were subsequently made as described previously (Cheng et al, 2021). Sequencing was performed at Novogene Ltd, UK, using an Illumina Novaseq 6000 with a paired-end 150 strategy. scRNAseq libraries were sequenced at an average depth of 300,000 reads/cells, and scATACseq libraries were sequenced at an average depth of 500,000 reads/cells.

## Smart-seq3 single-cell library preparation and sequencing of human oocytes

Single oocyte RNA-sequencing libraries were prepared using the Smart-seq3 protocol as previously described (Hagemann-Jensen et al, 2020). Resulting libraries were sequenced at Novogene Ltd., UK, using an Illumina Novaseq 6000 with a paired-end 150 strategy targeting an average sequencing depth of 300,000 reads/cell.

## Sequencing data processing

Raw reads generated by Smart-seq3 were mapped using the zUMIs pipeline as previously described (Hagemann-Jensen et al, 2020; Parekh et al, 2018). Raw reads generated by scATAC libraries were mapped as previously described (Cheng et al, 2021; Lentini et al, 2022).

## RNA isolation

Snap-frozen placental samples ($n = 25$) were lyzed in TRI Reagent (Sigma-Aldrich) using the RETCH MM 400 Mixer Mill (Fisher Scientific) for $4 \times 15$ s at 25 Hz. The samples were then incubated for 5 min at RT before 140 μl chloroform was added. Following incubation for 3 min at RT and centrifugation at 4 °C for 15 min, 350 μL of the aqueous layer was retrieved and mixed 1:1 with isopropanol. Purification of the RNA was performed using the ReliaPrep RNA Miniprep Systems (Promega). The manufacturer's protocol was used with the following modifications: all centrifugations of 15 s and 30 s were extended to 1 min and the samples were centrifuged for 1 min at full speed before elution of the RNA. The RNA was eluted in 30–50 μl Nuclease free water and quantified with Nanodrop 1000 spectrophotometer (ThermoFisher Scientific). RNA quality was assessed using the Qubit RNA IQ assay kit (Thermo Fisher Scientific) according to the manufacturer's protocol. Only samples with RNA IQ values > 8, representing good RNA quality, were used for RNA sequencing library preparation.

## Library preparation and bulk RNA sequencing

Libraries for bulk RNA sequencing were prepared using the prime-seq protocol as previously described (Janjic et al, 2022). In brief, reverse transcription was performed using 40 ng RNA per sample, and Exonuclease I was added for digestion of residual primers. cDNA pre-amplification and second-strand synthesis was performed using a PCR machine (Techtum). For cDNA quantity assessment, the Qubit dsDNA HS Assay Kit (Thermo Fisher Scientific) was used according to the manufacturer's instructions. The quality of the cDNA was evaluated using the Bioanalyzer High Sensitivity DNA kit (Agilent), according to protocol.

The sequencing library was constructed using the NEBNext Ultra II FS Kit (New England Biolabs) according to the prime-seq protocol. Briefly, the cDNA was fragmented, and adapters were ligated. Size-selection was performed using SPRI beads and the selected fragments were amplified through PCR. Following amplification, a second round of size-selection was performed. The quality and quantity of the library were assessed using the High Sensitivity DNA Bioanalyzer Chip. The samples were sequenced on Illumina NovaSeq 6000 at an average depth of 10 million reads per sample.

The raw data from bulk RNA sequencing was quality controlled using FastQC (version 0.11.9). Trimming of poly(A) tails and filtering was performed using Cutadapt (version 4.1) and the zUMIs pipeline (Parekh et al, 2018) (version 2.9.7), respectively. STAR (Dobin et al, 2013) (version 2.7.10a) was used for mapping of the filtered data to the mouse genome and the reads were counted using RSubread (Liao et al, 2019) (version 2.12.0).

## Data analysis

Data analysis was performed in R, version 4.3.0.

scRNAseq data was analyzed using Seurat version 4.3.0 (Hao et al, 2021). Cells with <500 features and >15% mitochondrial gene count (mouse oocyte) or <1000 features and >25% mitochondrial gene count (human oocyte) were filtered away. Differential expression was calculated using MAST version 1.25.1 (Finak et al, 2015). Differentially expressed genes were defined by adjusted *p*-value < 0.05 and log2 fold change >1 or <−1.

For re-analysis of data from Chen et al, DESeq2 version 1.40.1 was used for differential expression testing (Love et al, 2014). Differentially expressed genes were defined by adjusted *p*-value < 0.05 and log2 fold change >1 or <−1.

For bulk RNA-seq analysis, the quality of the samples was checked using Seurat version 4.3.0. Samples with $<2 \times 10^6$ counts ($n = 3$) were excluded from further analysis. Differential gene expression analysis was performed using DESeq2 version 1.40.1. Genes with a low expression (summed gene count ≤50) were removed. Differentially expressed genes were defined by adjusted *p*-value < 0.05 and log2 fold change >0.5 or <−0.5. Whole gene lists obtained from DESeq2 analysis were used for gene set enrichment analysis (GSEA) using the clusterProfiler package (v.4.8.1) and curated hallmark gene sets from the Molecular Signatures Database (MSigDB, v.2023.1) (Wu et al, 2021; Subramanian et al, 2005).

Pseudobulk sampling was done from mouse oocyte data by summarizing gene counts from all oocytes obtained from a single mouse. Pseudobulk data from this experiment was then integrated with data from Chen et al (Chen et al, 2022) using ComBat-Seq (Zhang et al, 2020). Two samples from the contemporary study were removed after QC of pseudobulk samples due to lower amounts of features and counts detected, making them outliers.

For the joint scATAC-seq data, raw sequence reads were trimmed based on quality (phred-scaled value of >20) and the presence of illumina adapters, and then aligned to the mm10 genome build using bowtie2 (2.3.5.2). Reads that were not mapped, not primary alignment, missing a mate, mapq <10, or overlapping ENCODEs blacklist regions (Amemiya et al, 2019) were removed.

scATAC data was analyzed using ArchR version 1.0.2 (Granja et al, 2021). Cells with TSS enrichment <0.5 and <1000 fragments were excluded from downstream analysis. Differentially accessible peaks were defined as FDR < 0.10 and log2 fold change >0.5 or <−0.5.

Data visualization was performed using ggplot2 version 3.4.2 and EnhancedVolcano version 1.18.0.

## Tissue sectioning

After collection, placentas were fixed in 4% PFA for 24 h before dehydration and embedding. Paraffin-embedded and formalin-fixed placentas were cut in 5 μM sections using a waterfall microtome (HM360, Microm). The sections were stretched in a 37–41 °C water bath, transferred to a SuperFrost Microscope Slide (Thermo Fisher) and dried at 40 °C for 1–2 h and at RT overnight.

## Hematoxylin and erythrosine staining

For morphological analysis of placentas using hematoxylin and erythrosine staining, tissue sections were deparaffinized in xylene and rehydrated using graded ethanol. Tissues were then stained with Mayer's hematoxylin for 6 min and differentiation with acid ethanol (70% EtOH, 1% HCl). Nuclear blueing was subsequently performed using Scott's tap water ($H_2O$, (mM $NaHCO_3$, 83 mM

MgSO$_4$) for 1 min before counterstaining using 0.3% erythrosine for 6 min. Afterwards, samples were dehydrated in graded ethanol and cleared in xylene, before mounting using pertex.

## Immunofluorescence staining

For immunofluorescence staining, tissue sections were deparaffined in xylene for $2 \times 15$ min and rehydrated using graded ethanol. Antigen retrieval was performed in microwave at pH 6 using Citrate Buffer (Sigma) for 5 min at 900 w. To reduce non-specific background staining, the samples were blocked with 10% bovine serum albumin (BSA) (for secondary antibodies produced in goat) or 5% donkey serum (for secondary antibodies produced in donkey) for 30 min at RT. Each sample was stained with two combinations of primary antibodies: firstly, anti-MCT1 (chicken anti-mouse, 1:200, Sigma, ab1286-I) and anti-pimonidazole conjugated with red fluorophore (mouse anti-pimonidazole, 1:100, Hydroxyprobe, hp11) and secondly, anti-CD31 (donkey anti-mouse, 1:100, Dianova, DIA-310) and anti-MCT4 (donkey anti-mouse, 1:200, Sigma, ab3314P). Staining was performed overnight at 4 °C. Secondary antibodies were then added, and all secondary antibodies were diluted 1:500. The samples were incubated for 1 h and quenching of autofluorescence was thereafter performed with 0.1% Sudan Black-B (Sigma) for 10 min. For nuclear staining, the samples were incubated with DAPI (1:5000) for 3 min. Mounting was performed using ProLong Gold Antifade (Thermo Fisher Scientific) mounting media and the samples were stored at RT for 24 h and at 4 °C thereafter until imaging. Unless otherwise stated, all incubations were performed at RT in a humidity chamber.

Imaging of the samples was performed using the LSM 700 microscope (Zeiss) using Z-stack imaging. A total of at least three images (technical replicates) were acquired for each sample.

## Image analysis

For analysis of placental sections stained for CD31 and MCT4, CD31-positive (CD31+) cells in the junctional zone and decidua were manually quantified using Fiji for assessment of the placental angiogenesis. A maximum projection image was created for each image for enhanced contrast. Identification of CD31+ cells is based on the staining intensity and the shape of the staining (vessel-like morphology). The nuclei surrounded by or in immediate vicinity of CD31+ staining were selected and counted as positive. At least 3 images per slide were quantified.

To quantify the number of nuclei in the junctional zone and decidua, CellProfiler (Stirling et al, 2021) (version 4.2.4) was used. The labyrinth zone was identified and removed from each image using the MCT4 staining as a marker. The IdentifyPrimaryObject module was used to keep objects with a diameter between 10–50 pixels, representing the MCT4-stained vessels in the labyrinth zone. Then, to fill holes between the selected objects, each object was expanded with 10 pixels using the ExpandOrShrinkObject function. The MaskImage module was used to apply the expanded labyrinth objects to the DAPI channel to remove the labyrinth nuclei. Subsequently, the remaining nuclei were identified and quantified by applying the IdentifyPrimaryObject function to the DAPI channel. To filter out objects of improper sizes not reflecting true nuclei, objects with a diameter <3 or >30 pixels were removed from the analysis. The minimum cross entropy threshold was applied and clumped objects were distinguished by shape and separated based on intensity. The average proportion of CD31+ cells in the junctional zone and decidua was estimated for every image by calculating the ratio of CD31+ cells to the total number of nuclei.

For analysis of the level of hypoxia in the placenta, the total intensity of pimonidazole staining was quantified in Fiji and divided to the total area of the junctional zone and decidua, excluding the labyrinth zone area. The MCT1 staining was used for identification of the labyrinth zone.

At least 3 images for each sample (technical replicates) were obtained and analyzed. Analysis of pimonidazole and CD31 images was blinded.

## Statistical analysis

Statistical analysis of the results was performed in GraphPad prism (version 9). The normality of the data was evaluated using the Kolmogorov-Smirnov test and all data analyzed was found to be normally distributed. ROUT's test was performed for identification of statistical outliers and differences between two groups were assessed using the two-sided Student's t-test. Two-way ANOVA was used to assess the differences between groups when time was a second factor (OGTT, insulin measurements, glucose dynamics). ANCOVA was applied for analyses on the embryos, CRL, placental weight and the placental efficiency to correct for the effect caused by maternal litter size. One-way ANOVA with Bonferroni's multiple comparisons test was used to assess the differences between IVM-cultured oocytes. $P < 0.05$ was considered statistically significant. Estimation of sample sizes was done based on previous studies investigation embryonic phenotypes (Risal et al, 2019) and oocyte analysis (Risal et al, 2019; Chen et al, 2022).

## Study approval

The oocyte retrieval part of this study was conducted according to the guidelines of the Declaration of Helsinki and approved by the Swedish Ethics Authority. The ethical approval for this study was granted by the Regional Ethics Committee of Stockholm (Dnr 2010/549-31/2 Amendment 2012/66-32). Informed consent was obtained from all subjects involved in the study.

All animal experiments were approved by the Stockholm Ethical Committee for Animal Research (17538-2020 with amendment 18959-2021 and 8639-2022) in accordance with the legal requirements of the European Community (SJVFS 2017:40) and the directive 2010/63/EU of the European Parliament on the protection of animals used for scientific purposes. Animal care and procedures were performed in accordance with guidelines specified by European Council Directive and controlled by Comparative Medicine Biomedicum, Karolinska Institutet, Stockholm, Sweden.

## Graphics

The graphical abstract and illustration in Fig. 1 were created using BioRender.com.

## Data availability

The raw sequence data for mouse oocytes and placenta reported in this paper have been deposited in under the Bioproject accession no. PRJNA1001949. Other data including human oocyte raw sequence data is available from the authors upon reasonable request. Codes that support our results and figures will be available at GitHub upon publication: www.github.com/Denglab-KI.

## Peer review information

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

inducible factor-1 contributes to increased mitochondrial reactive oxygen species production in diabetes. eLife 11:e70714

## Acknowledgements

We thank Han-Pin Pui, Christina An Binh Nordentoft, Sara Torstensson and Haojiang Lu for technical assistance for in vivo studies, and Jian Zhao for technical assistance for microscopy. QD is Wallenberg Academy Fellow in Medicine and is supported by the Swedish Medical Research Council (no. 2018-02557 and 2020-00253), and faculty funding at Karolinska Institutet, Barndiabetesfonden (The Swedish Children's Diabetes Foundation), Diabetesfonden (The Swedish Diabetes Foundation). AZ is supported by KID funding from Karolinska Institutet, HJ is supported by Chinese Scholarship Council. We also thank the Histological Core Facility (Histocore) at Karolinska Institutet.

## Author contributions

**Allan Zhao**: Conceptualization; Data curation; Software; Formal analysis; Validation; Investigation; Visualization; Methodology; Writing—original draft; Project administration; Writing—review and editing. **Hong Jiang**: Data curation; Software; Formal analysis; Validation; Investigation; Visualization; Writing—original draft; Writing—review and editing. **Arturo Reyes Palomares**: Formal analysis; Investigation; Visualization; Methodology; Writing—review and editing. **Alice Larsson**: Formal analysis; Investigation; Visualization; Writing—review and editing. **Wenteng He**: Investigation; Methodology; Writing—review and editing. **Jacob Grünler**: Investigation; Methodology; Writing—review and editing. **Xiaowei Zheng**: Resources; Methodology; Writing—review and editing. **Kenny A Rodriguez-Wallberg**: Resources; Methodology; Writing—review and editing. **Sergiu-Bogdan Catrina**: Resources; Supervision; Methodology; Writing—review and editing. **Qiaolin Deng**: Conceptualization; Resources; Data curation; Software; Formal analysis; Supervision; Funding acquisition; Validation; Visualization; Methodology; Writing—original draft; Project administration; Writing—review and editing.

## Funding

## Disclosure and competing interests statement

The authors declare no competing interests.

# Expanded View Figures

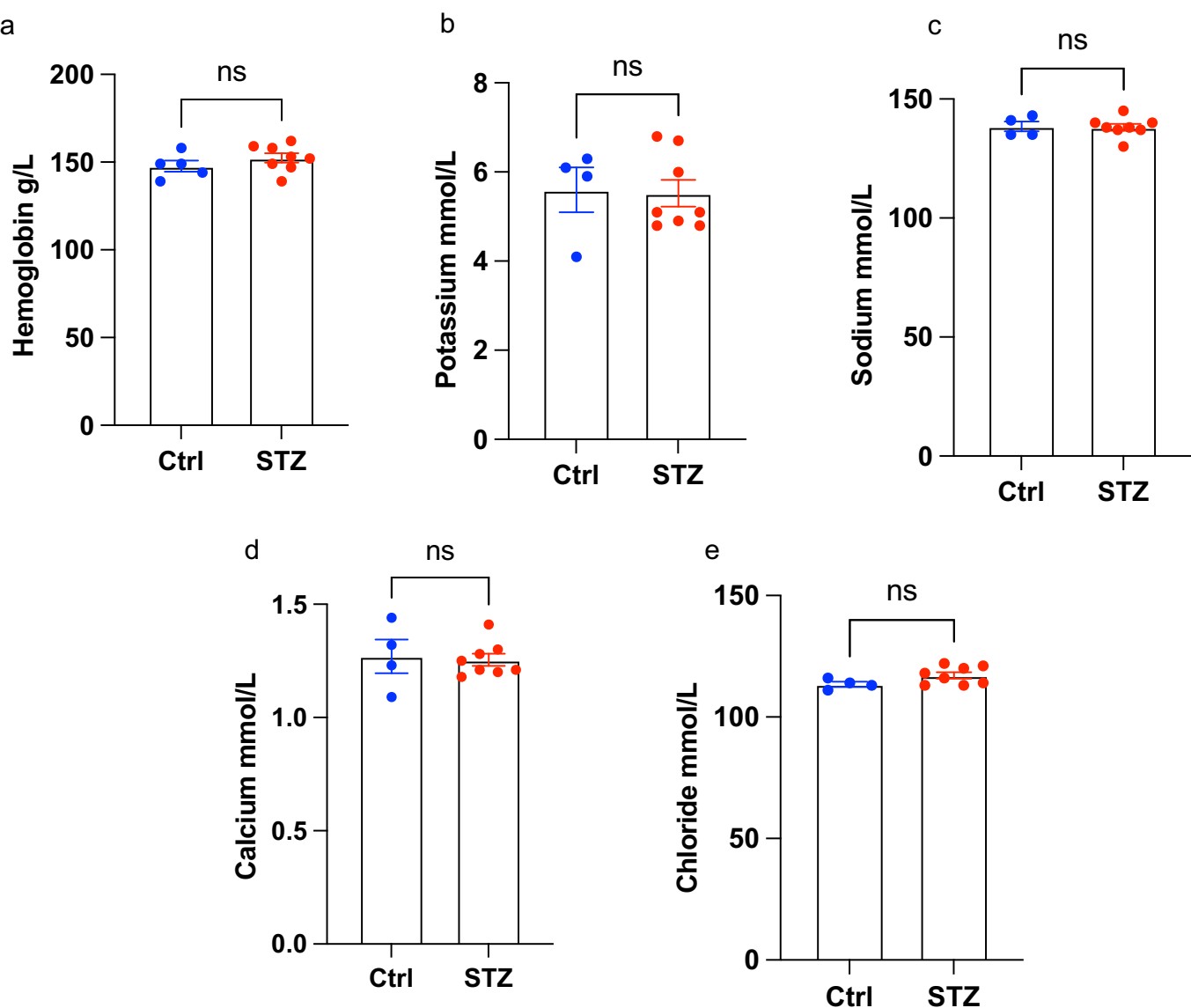

**Figure EV1.  Maternal mouse blood analysis.**

(**A–E**) Hemoglobin (*n* = 5 control mice, *n* = 8 STZ mice) (**A**), Potassium (*n* = 4 control mice, *n* = 8 STZ mice) (**B**), Sodium (*n* = 4 control mice, *n* = 8 STZ mice) (**C**), Calcium (*n* = 4 control mice, *n* = 8 STZ mice) (**D**) and Chloride (*n* = 4 control mice, *n* = 8 STZ mice) (**E**) levels in maternal control and STZ mice. STZ, Streptozotocin; ns, non-significant. Data information: Data depicted is biological replicates. In panels (**A–E**), data is presented as mean ± SEM, and was analyzed using Student's t-test.

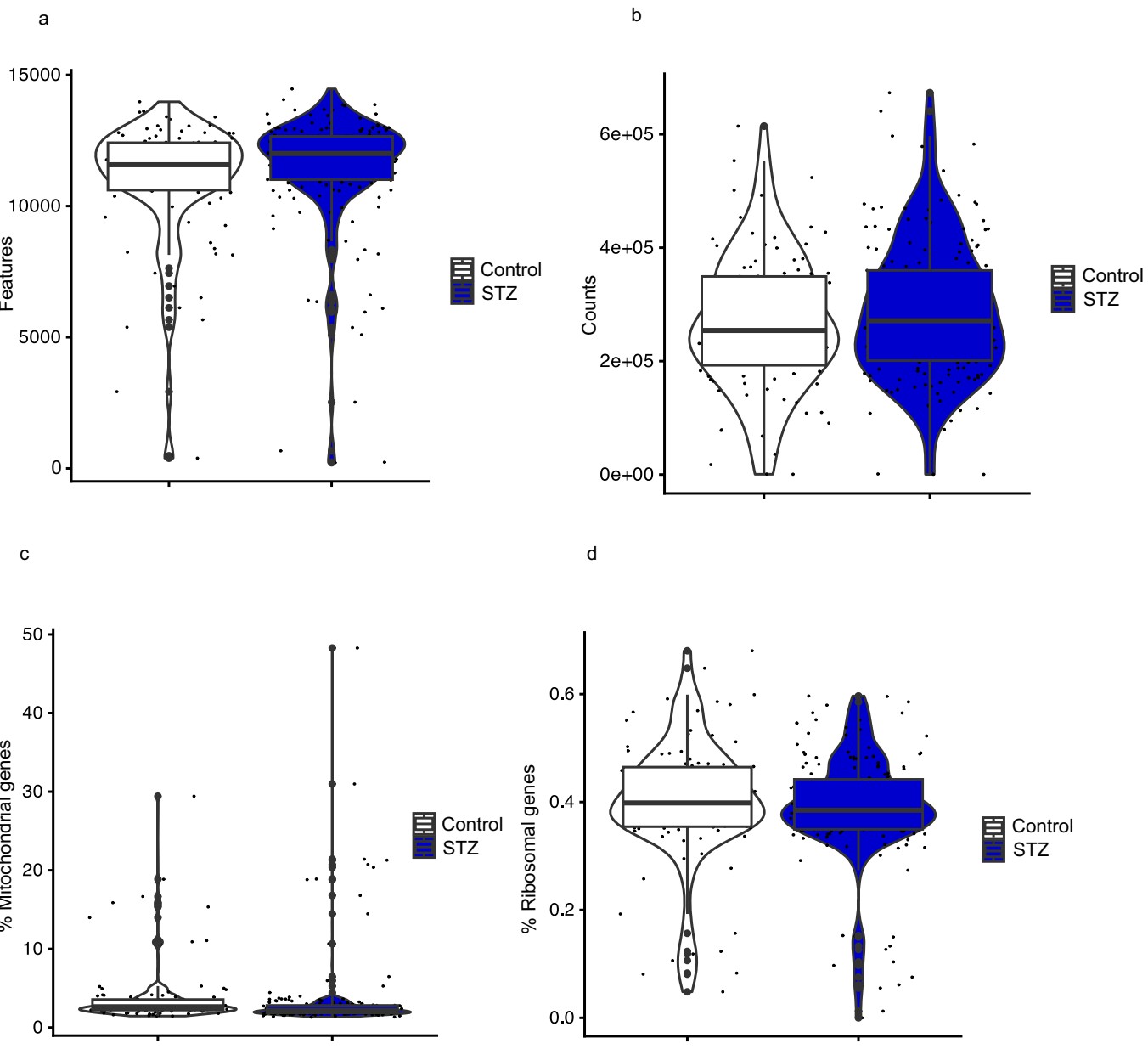

**Figure EV2. QC of mouse oocyte single-cell RNAseq data.**

(A) Amount of features (genes) per cell (5 control animals, $n = 100$ oocytes; 9 STZ animals, $n = 172$ oocytes). (B) Amount of counts per cell (5 control animals, $n = 100$ oocytes; 9 STZ animals, $n = 172$ oocytes). (C) Percentage of mitochondrial reads per cell (5 control animals, $n = 100$ oocytes; 9 STZ animals, $n = 172$ oocytes). (D) Amount of ribosomal reads per cell (5 control animals, $n = 100$ oocytes; 9 STZ animals, $n = 172$ oocytes). Data information: Data depicted is biological replicates and readcounts. For boxplots in panels (A–D), the lower and upper hinges of the box depict the 1st and 3rd quartiles, and the middle line depicts the median. The upper and lower whiskers extend to the largest or smallest value no further than 1.5 times the interquartile range (1.5*IQR) from the hinge. Data beyond the end of the whiskers are plotted as a larger dot, which are then the minimum and/or maximum values depicted. If no large dots are present, the whiskers extend to the minimum and/or maximum values.

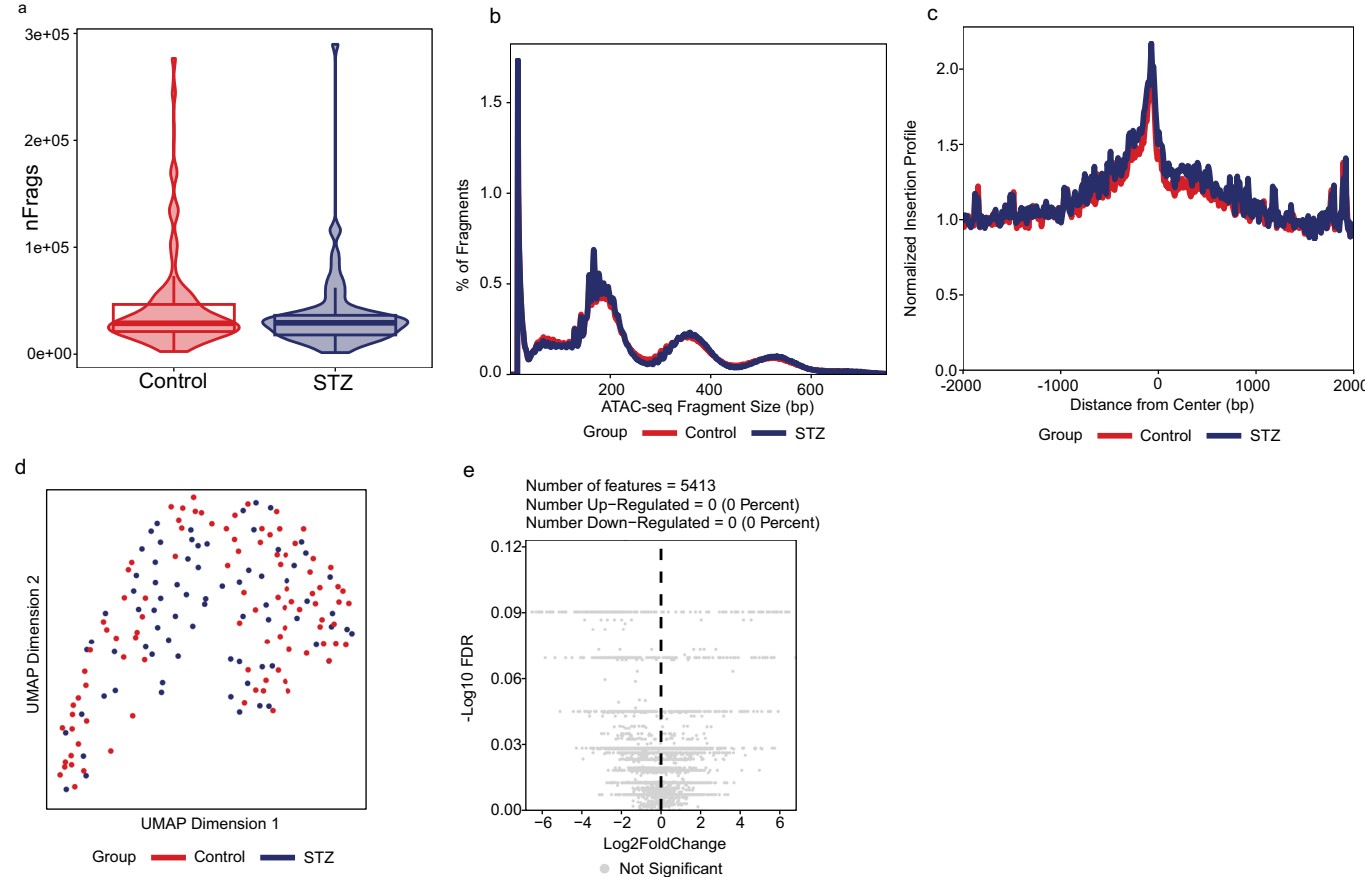

**Figure EV3. Mouse oocyte single-cell ATAC seq analysis.**

(**A**) Amount of unique fragments detected per cell from control ($n = 95$) and STZ oocytes ($n = 85$. (**B**) Fragment size distribution from control ($n = 95$) and STZ oocytes ($n = 85$ (**C**) TSS enrichment profile from control ($n = 95$) and STZ oocytes ($n = 85$). (**D**) UMAP depicting scATAC profiles from control ($n = 95$) and STZ oocytes ($n = 85$). (**E**) Volcano plot displaying no differentially accessible regions between control ($n = 85$) and STZ ($n = 95$) oocytes. TSS, Transcription starting site; UMAP, Uniform Manifold Approximation and Projection; STZ, Streptozotocin. Data information: Data depicted is biological replicates. For boxplots in panel (**A**), the lower and upper hinges of the box depict the 1st and 3rd quartiles, and the middle line depicts the median. The upper and lower whiskers extend to the largest or smallest value no further than 1.5 times the interquartile range (1.5*IQR) from the hinge. Data beyond the end of the whiskers are plotted as a violin plot, which are then the minimum and/or maximum values depicted.

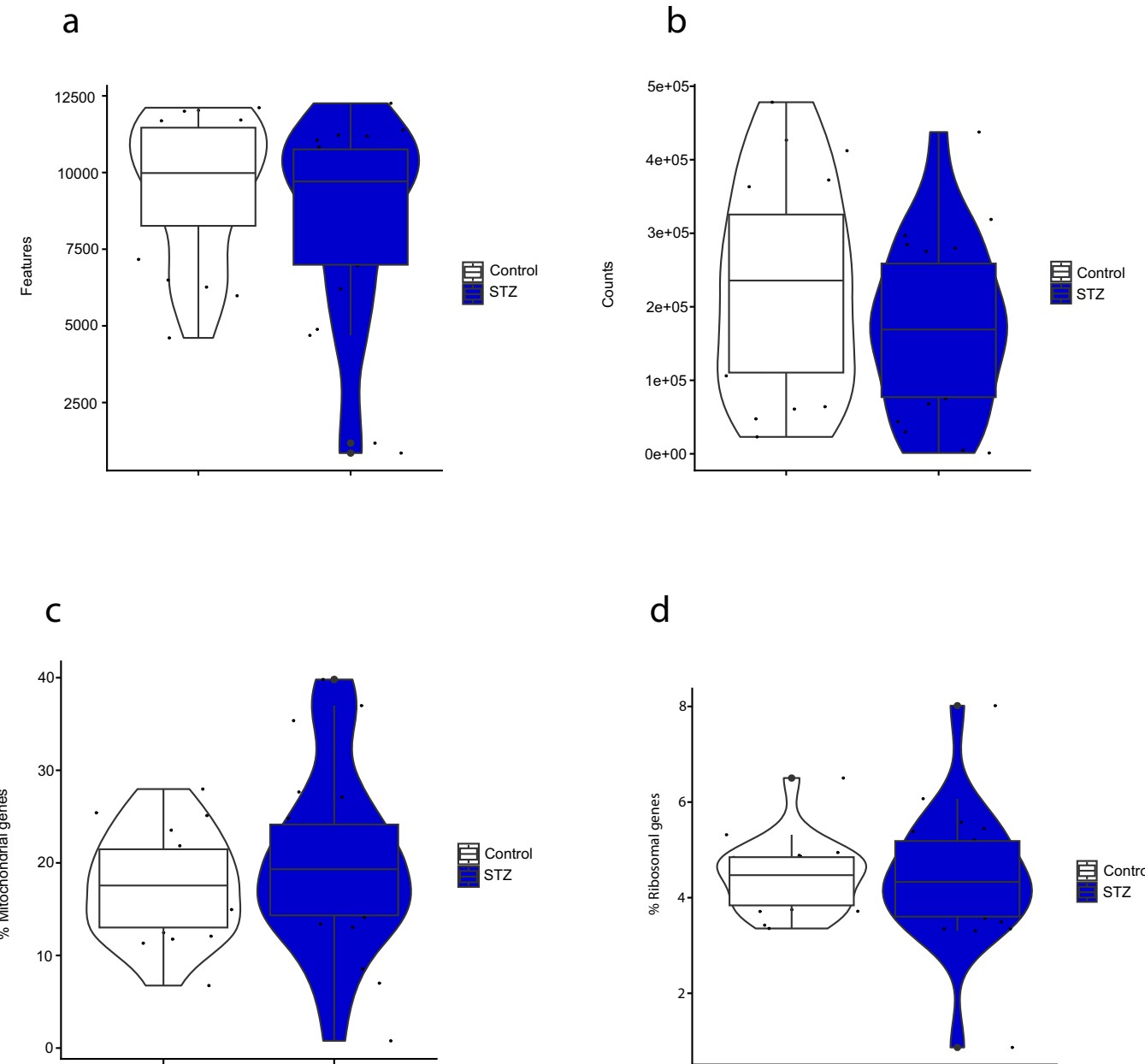

**Figure EV4.   QC of human oocyte single-cell RNAseq data.**

(**A**) Amount of features (genes) per cell (*n* = 20 for 2.5 mM, *n* = 21 for 5 mM, *n* = 20 for 10 mM). (**B**) Amount of counts per cell (*n* = 20 for 2.5 mM, *n* = 21 for 5 mM, *n* = 20 for 10 mM). (**C**) Percentage of mitochondrial reads per cell (*n* = 20 for 2.5 mM, *n* = 21 for 5 mM, *n* = 20 for 10 mM). (**D**) Amount of ribosomal reads per cell (*n* = 20 for 2.5 mM, *n* = 21 for 5 mM, *n* = 20 for 10 mM). Data information: Data depicted is biological replicates and readcounts. For boxplots in panels (**A–D**), the lower and upper hinges of the box depict the 1st and 3rd quartiles, and the middle line depicts the median. The upper and lower whiskers extend to the largest or smallest value no further than 1.5 times the interquartile range (1.5*IQR) from the hinge. Data beyond the end of the whiskers are plotted as a larger dot, which are then the minimum and/or maximum values depicted. If no large dots are present, the whiskers extend to the minimum and/or maximum values.

                                      