## [Peer Review File · EMBO Reports]

Appropriate glycemic management protects the germline but not uterine environment in hyperglycemia

Allan Zhao, Hong Jiang, Arturo Reyes Palomares, Alice Larsson, Wenteng He, Jacob Grünler, Xiaowei Zheng, Kenny Rodriguez-Wallberg, Sergiu-Bogdan Catrina, and Qiaolin Deng

Corresponding author(s): Qiaolin Deng (qiaolin.deng@ki.se)

Review Timeline:

Transfer Date:	21st Nov 23
Editorial Decision:	28th Nov 23
Appeal:	29th Nov 23
Editor's Response to Appeal:	30th Nov 23
Revision Received:	30th Nov 23
Editorial Decision:	26th Jan 24
Revision Received:	2nd Feb 24
Accepted:	8th Feb 24

Editor: Esther Schnapp

Transaction Report: This manuscript was transferred to EMBO reports following peer review at EMBO Molecular Medicine.

Date: 20th Oct 23 03:52:03

Last Sent: 20th Oct 23 03:52:03

Triggered By: Poonam Bheda

From: contact@embomolmed.org

To: qiaolin.deng@ki.se

Subject: EMM-2023-18672-V2-Q Decision Letter

Message: 19th Oct 2023

Decision on your manuscript EMM-2023-18672-V2-Q

Dear Dr. Deng,

Thank you for the submission of your manuscript "Appropriate glycemc management protects the germline but not uterine environment in type 1 diabetes". We have now received feedback from the three referees who had agreed to review your manuscript.

As you will see from the reports below, the referees acknowledge the potential interest of the study, however they also mention some shortcomings. In particular, Reviewer 1 finds the two models in the manuscript inappropriate as a model of appropriately controlled glycemia - rather that low-dose STZ in mouse more likely recapitulates modest T1D and the high-dose glucose treatment of oocytes was also not positively controlled as a proper model. In addition, Reviewer 3 also commented that expanding the study to include human oocytes directly from T1D cases would be beneficial to the study. Given these concerns and suggestions on the models used in the manuscript, I am afraid I see little choice but to return the manuscript to you at this point with the decision that we cannot offer to publish it.

While we cannot pursue this manuscript further, we encourage you to transfer your study to our not-for-profit open-access sister journal, Life Science Alliance (LSA). We shared your manuscript and the accompanying reviews with LSA Executive Editor, Eric Sawey, who is interested in these findings, and would like to invite further consideration of this manuscript at LSA pending the following revisions:

- Address Reviewer 1's comments, particularly those related to the model recapitulating modest cases of T1D.
- Address Reviewer 3's comments, except for Comment #2, unless that data is readily available.

We encourage you to use the link below to transfer your manuscript to LSA. You do not need to revise the manuscript before transferring it to LSA. Once you

transfer, Dr. Sawey will email you an invitation to revise and resubmit, listing the same revision requests as mentioned above. Please feel free to reach out at e.sawey@life-science-alliance.org if you have any questions about the LSA journal, the transfer process or the revisions requested.

I am very sorry to disappoint you on this occasion and I hope you will view the possibility of a transfer favorably. If this is the case, please use the link below to transfer the manuscript directly.

With kind regards,

Poonam Bheda

Poonam Bheda
Scientific Editor
EMBO Molecular Medicine

***** Reviewer's comments *****

Referee #1

The two models presented in this study are inadequate to support the claimed conclusions.

The mouse model produced in this study by injecting low doses of streptozotocin (STZ) represents modest cases of type I diabetes (T1D) - not T1D cases whose blood glucose is well controlled by pharmacological treatments. Production of STZ-induced T1D mouse model is a standard practice, and the claim that a low STZ dose produced modest T1D has little novelty.

The IVM culture of human oocytes collected from non-diabetic females in the presence of glucose higher than the standard (5 mM versus 2.5 mM) has little relevance to the T1D patients with controlled glucose level, and the negative outcomes (i.e., lack of morphological or transcriptomic changes) are presented without positive control.

Because the clinically very significant claims presented in this manuscript are not supported by adequate models, I think publication of this study, in its current form, may rather mislead the non-expert audience than help their understanding of reproductive impact of T1D.

Referee #1

In this study, 10-weeks-old female C57BL/6J mice were exposed to low doses of streptozotocin (STZ, 50 mg/kg) by injection - through unspecified route - over five consecutive days. In contrast to the authors' prior experience that the same STZ injection scheme caused severe diabetes in MALE mice, the exposed female mice showed signs of modest type I diabetes (T1D), including "slightly elevated and stable glucose level over the whole experimental period (Fig. 1C)" as well as abnormal OGTT and HbA1c. Different from the commonly used, severe T1D model mice produced by a high-dose and one-shot (150 mg/kg) STZ injection, the modest T1D model presented in this study may have some novelty although the authors do not explicitly state whether the "slight elevation" of blood glucose (Fig. 1C) is statistically significant or not (Fig. 1C does not show any asterisks or ns). The low-STZ T1D model did not affect body weight or estrous cyclicity or transcriptomes of the superovulated oocytes. However, the authors claim that the low-STZ T1D had negative impact to in utero development of fetuses with anomalies observed in placenta.

The authors repeatedly and explicitly emphasize that their low-STZ T1D mouse model resembles clinical human T1D cases whose blood glucose is appropriately managed or controlled. This is my major concern. The low-STZ T1D mouse model presented in this manuscript has never been treated to control their blood glucose level after its initiation of the T1D symptoms, and by mechanism it is a model of modest T1D case with no specific need of treatment. In contrast, human T1D patients whose blood glucose level is high receive pharmacological treatments, and their controlled blood glucose does not mean that their fundamental T1D state (i.e., the lack of endogenous insulin production by the beta cells) is improved. Because of this critical difference between the low-STZ T1D model and the actual T1D patients with controlled blood glucose level, it is misleading to claim that adequate management of blood glucose level prevents oocytes from transcriptional alterations based on the current mouse model. It may be possible to suggest that modest cases of T1D can suffer from fertility issues due to placental anomalies whereas health of their oocytes might be unaffected; however, changing the central claim of this manuscript in such an alternative story does not seem achievable by simply amending the title and text wording.

The authors claim that retrieval of oocytes from T1D patients for research purposes is ethically inappropriate because T1D patients with controlled blood glucose experience low rates of fertility issues. However, even if this claim is acceptable, IVM culture of human oocytes collected from non-diabetic females for IVF purposes in the presence of 2.5 mM or 5 mM glucose needs more justification. Specifically, this experiment lacks positive control - namely, the authors need to show transcriptomic or morphological changes in the IVM

culture oocytes maintained in a higher concentration of glucose. If oocytes are resistant to even very high glucose, then the conclusion will be different from the current manuscript that managed blood glucose of T1D patients may protect oocytes.

In the last paragraph of the Results section, the authors referred to Fig. 4h, which I believe an error and needs to be corrected to Fig. 5g.

The authors discuss that maternal T1D can still affect the uterine environment likely because the placenta is a richly infused organ with blood compared to the ovary. But I do not find adequately strong rationale supporting this speculation.

Referee #2

The study investigated the impact of maternal glycemic control in women with Type 1 Diabetes (T1D) on fetal development and growth outcomes. The findings suggest that even with proper management of glycemic levels, there are still risks associated with T1D pregnancies, such as birth defects, preterm birth, and fetal growth deviations. The study points to the uterine environment as a potentially significant factor in these adverse outcomes, with implications for the future health of the offspring.

The use of a maternal T1D mouse model and the administration of low-dose streptozotocin (STZ) to mimic the condition is an interesting approach. The data showing no changes in the transcriptome of oocytes exposed to managed glucose levels suggests that the impact might not be directly on the oocytes themselves. Instead, the study highlights the role of an adverse uterine environment and placental dysfunction in fetal growth deviations.

The study's emphasis on hypoxia conditions in the placenta and their link to fetal growth restriction is noteworthy. Hypoxia, or inadequate oxygen supply, is known to be a critical factor in various developmental and health outcomes, and understanding the pathways affected by maternal hypoxia in the placenta could provide valuable insights.

It's a valid concern that the study should investigate how hypoxia conditions affect fetal growth and identify specific pathways affected by maternal hypoxia in the fetus. This would help provide a more comprehensive understanding of the mechanisms involved in adverse outcomes in T1D pregnancies and might lead to potential interventions or therapies to mitigate these risks.

Overall, this research appears to be making important contributions to our understanding of diabetic embryopathy and the role of the uterine environment

in fetal development, with the suggestion of hypoxia conditions as a critical element. Further investigation into the specific pathways affected by maternal hypoxia could be a valuable next step in this area of study.

Referee #3

In this paper, authors aimed to investigate the impact of appropriately managed glycemic levels in maternal type 1 diabetes (T1D) on oocyte transcriptome, chromatin accessibility, intrauterine development, and placental function. The results revealed that appropriately managed maternal glycemic levels preserved the oocyte transcriptome and chromatin accessibility, both in mice and human oocytes. However, fetal growth and placental function were still adversely affected despite glycemic control, highlighting the importance of the uterine environment in developmental programming. Placental dysfunction, characterized by increased angiogenesis and hypoxia, was identified as a potential contributing factor to fetal growth deviations in the context of appropriately managed maternal T1D. These findings emphasize the significance of achieving proper pregestational glycemic control and the need for further research on therapeutic interventions during pregnancy to mitigate adverse effects on fetal development.

Comments

- 1) A detailed discussion comparing this study's design and findings with those of Chen et al. could provide valuable insights into the comparison of transcriptomic profiles among control, appropriately managed, and poorly managed glycemia in oocyte transcriptomes.
- 2) Expanding the study to include human oocytes from type 1 diabetes cases and controls, rather than creating a diabetic environment using 5mM glucose, would enhance the study's relevance. Addressing potential biases arising from comparing transcriptomic profiles of non-diabetic cells cultured in control versus diabetic environments, rather than studying diabetic versus non-diabetic oocytes, should be discussed.
- 3) The paper should include a discussion of its limitations and potential biases to ensure a more accurate interpretation of the results.

As a service to authors, EMBO provides authors with the possibility to transfer a manuscript that one journal cannot offer to publish to another EMBO publication. The full manuscript and if applicable, reviewers reports are automatically sent to the receiving journal to allow for fast handling and a prompt decision on your manuscript. For more details of this service, and to transfer your manuscript to another EMBO title please click on *Link Unavailable*

**Karolinska
Institutet**

Dear scientific editor of *EMBO Molecular Medicine*, Dr. Poonam Bheda:

It is difficult for us to understand the decision based on three reviewers' comments provided below. To our interpretation, the reviewers #2 and #3 are quite positive. We can address reviewer #3's comments in the revised manuscript and potentially new data. We are also very encouraged to see there are few questions raised about results and data analyses.

Therefore, it is concerning if the decision is mainly based on reviewer#1's comments, in which the major criticism is that our model does not completely mimic T1D patients due to lack of exogenous treatment with insulin. However, this is a common challenge to many mouse models that often only represent certain disease features. Our model indeed recapitulates the glycemic levels and fluctuation often achieved in appropriately managed patients with T1D. It is a significant step forward to study the pathological effects of such glycemic levels without other confounding factors due to 10-weeks maintenance which is lacking in current studies. Our study provided insights on the unsolved issue why women with appropriately managed glycemia continue to experience a higher incidence of pregnancy complications. However, we do need to stress more about the limitation of our model.

Due to the conclusions from the recent publication in *Nature* (<https://www.nature.com/articles/s41586-022-04756-4>), it is highly relevant to the field to highlight the importance of pregestational glycemic control and alternative mechanism of maternal epigenetic inheritance. We feel strong responsibility to reach the public about these findings and *EMBO Molecular Medicine* is a highly respected channel for this. We do agree that we should discuss more about the limitation of our mouse model compared with T1D patients. Hence, we would like to provide a detailed response to the reviewers' comments as outlined below. We sincerely hope that you can reconsider the decision.

Yours sincerely,

Qiaolin Deng, PhD, Associate professor
Dept. Physiology and Pharmacology
Karolinska Institutet
Center of Molecular Medicine
Karolinska University Hospital

Referee #1

The two models presented in this study are inadequate to support the claimed conclusions.

The mouse model produced in this study by injecting low doses of streptozotocin (STZ) represents modest cases of type I diabetes (T1D) - not T1D cases whose blood glucose is well controlled by pharmacological treatments. Production of STZ-induced T1D mouse model is a standard practice, and the claim that a low STZ dose produced modest T1D has little novelty.

We thank the reviewer for the comment about our mouse model. We do agree that there is a difference between our T1D model and appropriately managed patients of T1D with treatment. However, in this study we are focusing on the pathological effects of the glycemic level often achieved in appropriately managed patients with T1D as so far, no study is addressing these questions. We are doing so since hyperglycemia is thought to be the important factor central for development of diabetic complications in T1D and all other forms of diabetes. Patients with appropriately managed glycemic levels are increasingly common but little research is done on this patient group.

While we agree that a low dose of STZ has been used to produce modest T1D, the critical difference is that our mouse model can exclude any potential STZ-related toxicity or confounding effects in follow-up studies as we are able to maintain females largely healthy up to 10 weeks before the following experiments. Our study is also the first to carefully phenotype these T1D female mice in major reproductive and metabolic functions. Therefore, the novelty lies in our experimental design in studying pathological effects of the glycemic level often achieved in appropriately managed patients with T1D, and how such glycemic levels affects the germline and uterine environment respectively.

The IVM culture of human oocytes collected from non-diabetic females in the presence of glucose higher than the standard (5 mM versus 2.5 mM) has little relevance to the T1D patients with controlled glucose level, and the negative outcomes (i.e., lack of morphological or transcriptomic changes) are presented without positive control.

We thank the reviewer for this comment about our experiment investigating the oocyte transcriptome. The choice of glucose concentration is guided by the previous study from Chen et al., (Nature, 605, 761-766, 2022) suggesting that performing IVM in 10 or 15mM affects the *TET3* expression in oocytes in contrast to 2.5mM (Figure 1A). We therefore investigated a lower concentration of glucose, which is still significantly higher than the control level, mimicking a more appropriately managed disease state in terms of glycemic levels. Also shown by Chen et al. this glycemic level (5mM) is the average concentration in the follicular fluid measured in patients with diabetes (Figure 1B).

Figure 1. Excerpt of Extended Data Fig.6 from paper by Chen et al., *Maternal inheritance of glucose intolerance via oocyte *TET3* insufficiency*, Nature, 605, 761-766 (2022). (A) *TET3* mRNA expression in human MII oocytes from IVM under the indicated glucose concentrations. For the 2.5mM, 10mM and 15mM groups, $n=9$, 9 and 11 oocytes, respectively. (B) Human follicular fluid glucose concentrations measured in the patients with or without diabetes. *h*,

We have, in addition, also cultured human oocytes at a higher concentration (10 mM) as a positive control. At this concentration, we still have not observed morphological changes during maturation. However, we do see a significant effect on the oocyte transcriptome, as the number of genes detected and the number of counts per cell are significantly lower for the 10mM group, whilst there is no difference between the 2.5mM and 5mM groups (Figure 2). We therefore believe that a more severe increase in glycemic levels during IVM will impact the overall oocyte transcriptome quality, further indicating the importance of proper pregestational control in maternal diabetes.

Figure 2. Number of counts (UMIs) (a) and genes/features detected (b) per cell for MII oocytes after IVM in 2.5, 5 and 10mM glucose. Significance is tested using one-way ANOVA with Bonferroni's post-hoc test.

Because the clinically very significant claims presented in this manuscript are not supported by adequate models, I think publication of this study, in its current form, may rather mislead the non-expert audience than help their understanding of reproductive impact of T1D.

We believe that the major clinical significance from our results lies in how we study the pathological effects of the glycemic level often achieved in appropriately managed patients with T1D, as the glycemic level and hyperglycemia is one of the most important factors when it comes to development of diabetes complications. It is true that our model does not perfectly reflect T1D patients with appropriate glycemic levels with pharmacological treatment, but instead mainly focuses on one critical disease feature (i.e. glycemic levels). It is indeed a challenge common for most other mouse models to perfectly mimic diseases, and ours is no exception. However, we believe that our model is sufficiently similar to the patients with diabetes when studying the effect of glycemic levels often seen in appropriately managed T1D patients. We do apologize that we fail to convey the limitation to our model to the reviewer#1 and agree that we cannot publish it in its current form without further clarification and revision. We will do that in our revision to strengthen our message.

Referee #1

In this study, 10-weeks-old female C57BL/6J mice were exposed to low doses of streptozotocin (STZ, 50 mg/kg) by injection - through unspecified route - over five consecutive days. In contrast to the authors' prior experience that the same STZ injection scheme caused severe diabetes in MALE mice, the exposed female mice showed signs of modest type I diabetes (T1D), including "slightly elevated and stable glucose level over the whole experimental period (Fig. 1C)" as well as abnormal OGTT and HbA1c. Different from the commonly used, severe T1D model mice produced by a high-dose and one-shot (150 mg/kg) STZ injection, the modest T1D model presented in this study may have some novelty although the authors do not explicitly state whether the "slight elevation" of blood glucose (Fig. 1C) is statistically significant or not (Fig. 1C does not show any asterisks or ns). The low-STZ T1D model did not affect body weight or estrous cyclicity or transcriptomes of the superovulated oocytes. However, the authors claim that the low-STZ T1D had negative impact to in utero development of fetuses with anomalies observed in placenta.

We thank the reviewer for this comment. We apologize that we have not mentioned the administration route, which is intraperitoneal. We also apologize that the star displaying significance in Fig. 1C has been misplaced in the figure, and instead is above the two data points for week 10. This elevation is significantly different as tested by a two-way ANOVA. Below is the revised Figure 1C.

The authors repeatedly and explicitly emphasize that their low-STZ T1D mouse model resembles clinical human T1D cases whose blood glucose is appropriately managed or controlled. This is my major concern. The low-STZ T1D mouse model presented in this manuscript has never been treated to control their blood glucose level after its initiation of the T1D symptoms, and by mechanism it is a model of modest T1D case with no specific need of treatment. In contrast, human T1D patients whose blood glucose level is high receive pharmacological treatments, and their controlled blood glucose does not mean that their fundamental T1D state (i.e., the lack of endogenous insulin production by the beta cells) is improved. Because of this critical difference between the low-STZ T1D model and the actual T1D patients with controlled blood glucose level, it is misleading to claim that adequate management of blood glucose level prevents oocytes from transcriptional alterations based on the current mouse model. It may be possible to suggest that modest cases of T1D can suffer from fertility issues due to placental anomalies whereas health of their oocytes might be unaffected; however, changing the central claim of this manuscript in such an alternative story does not seem achievable by simply amending the title and text wording.

We agree that a main difference between our mouse model and T1D patients with appropriate glycemic control is the lack of exogenous insulin supplementation, which is a limitation of our model as well as many mouse models of diabetes. However, as stated and elaborated on before, in this study we are focusing on the glycemic level as the pathological factor, since hyperglycemia is thought to be the crucial factor underlying development of diabetic complications. We therefore believe that our mouse model is adequate for studying the glycemic levels seen in appropriately managed T1D, as our model reflects similar glycemic levels. Hence, the message we want to convey is that glycemic levels corresponding to appropriate glycemic management in diabetes, which is more and more common in clinical cases (both T1D but also T2D), do not affect the oocyte transcriptome or chromatin accessibility but does affect the uterine environment and fetal development. We think our claim is scientifically sound but agree that the limitation of the model in comparison to T1D patients' treatment should be furthermore discussed to clarify any potential misunderstanding.

The authors claim that retrieval of oocytes from T1D patients for research purposes is ethically inappropriate because T1D patients with controlled blood glucose experience low rates of fertility issues. However, even if this claim is acceptable, IVM culture of human oocytes collected from non-diabetic females for IVF purposes in the presence of 2.5 mM or 5 mM glucose needs more justification. Specifically, this experiment lacks positive control - namely, the authors need to show transcriptomic or morphological changes in the IVM culture oocytes maintained in a higher concentration of glucose. If oocytes are resistant to even very high glucose, then the conclusion will be different from the current manuscript that managed blood glucose of T1D patients may protect oocytes.

This is the same question asked in the Novelty/Model system part by the Reviewer#1. Please also refer to the answer there. In short, we have based our culture condition on previous findings that extreme culture conditions of 10 and 15mM glucose induced transcriptomic changes during IVM, inducing *TET3* to potentiate glucose intolerance in offspring. Moreover, as written in the manuscript, we have chosen 5mM as the diabetic condition due to the finding that this is an average level in the follicular fluid of patients with diabetes. We therefore believe that these levels are proper to be tested in IVM. We now also

provided new results of 10mM glucose treatment on oocytes as a positive control, in which we see large effects on the overall oocyte transcriptome quality.

In the last paragraph of the Results section, the authors referred to Fig. 4h, which I believe an error and needs to be corrected to Fig. 5g.

We apologize for this mistake and have corrected it to Fig. 5g.

The authors discuss that maternal T1D can still affect the uterine environment likely because the placenta is a richly infused organ with blood compared to the ovary. But I do not find adequately strong rationale supporting this speculation.

We thank the reviewer for this comment and will expand our discussion about this topic.

Referee #2

The study investigated the impact of maternal glycemic control in women with Type 1 Diabetes (T1D) on fetal development and growth outcomes. The findings suggest that even with proper management of glycemic levels, there are still risks associated with T1D pregnancies, such as birth defects, preterm birth, and fetal growth deviations. The study points to the uterine environment as a potentially significant factor in these adverse outcomes, with implications for the future health of the offspring.

The use of a maternal T1D mouse model and the administration of low-dose streptozotocin (STZ) to mimic the condition is an interesting approach. The data showing no changes in the transcriptome of oocytes exposed to managed glucose levels suggests that the impact might not be directly on the oocytes themselves. Instead, the study highlights the role of an adverse uterine environment and placental dysfunction in fetal growth deviations.

The study's emphasis on hypoxia conditions in the placenta and their link to fetal growth restriction is noteworthy. Hypoxia, or inadequate oxygen supply, is known to be a critical factor in various developmental and health outcomes, and understanding the pathways affected by maternal hypoxia in the placenta could provide valuable insights.

It's a valid concern that the study should investigate how hypoxia conditions affect fetal growth and identify specific pathways affected by maternal hypoxia in the fetus. This would help provide a more comprehensive understanding of the mechanisms involved in adverse outcomes in T1D pregnancies and might lead to potential interventions or therapies to mitigate these risks.

Overall, this research appears to be making important contributions to our understanding of diabetic embryopathy and the role of the uterine environment in fetal development, with the suggestion of hypoxia conditions as a critical element. Further investigation into the specific pathways affected by maternal hypoxia could be a valuable next step in this area of study.

We thank the reviewer for these positive remarks.

Referee #3

In this paper, authors aimed to investigate the impact of appropriately managed glycemic levels in maternal type 1 diabetes (T1D) on oocyte transcriptome, chromatin accessibility, intrauterine development, and placental function. The results revealed that appropriately managed maternal glycemic levels preserved the oocyte transcriptome and chromatin accessibility, both in mice and human oocytes. However, fetal growth and placental function were still adversely affected despite glycemic control, highlighting the importance of the uterine environment in developmental programming. Placental dysfunction, characterized by increased angiogenesis and hypoxia, was identified as a potential contributing factor to fetal growth deviations in the context of appropriately managed maternal T1D. These findings emphasize the significance of achieving proper pregestational glycemic control and the need for further research on therapeutic interventions during pregnancy to mitigate adverse effects on fetal development.

Comments

1) A detailed discussion comparing this study's design and findings with those of Chen et al. could provide valuable insights into the comparison of transcriptomic profiles among control, appropriately managed, and poorly managed glycemia in oocyte transcriptomes.

We thank the reviewer for this comment and agree that further discussion about the study design would be important. We will add this in a revised version of the manuscript.

2) Expanding the study to include human oocytes from type 1 diabetes cases and controls, rather than creating a diabetic environment using 5mM glucose, would enhance the study's relevance. Addressing potential biases arising from comparing transcriptomic profiles of non-diabetic cells cultured in control versus diabetic environments, rather than studying diabetic versus non-diabetic oocytes, should be discussed.

We agree with the reviewer that adding human oocytes from T1D patients would be the best way of investigating the oocyte transcriptome in these patients. However, as described in the manuscript, collection of oocytes from T1D patients solely for research purposes is ethically inappropriate in Sweden, and as few of these patients need fertility treatment, it is also very difficult to obtain these samples in the IVF clinic. Therefore, we chose to use the strategy described in the paper. We do, however, agree that increased discussion of how this model system might not be completely reflective of the in vivo oocyte maturation process is suitable, and this will be added in a revised manuscript.

3) The paper should include a discussion of its limitations and potential biases to ensure a more accurate interpretation of the results.

We agree with the reviewer that a section of limitation and potential biases in the discussion is needed for further clarification to strengthen the manuscript. We will include this in a revised manuscript.

2nd EMM Decision Letter

Date: 15th Nov 23 11:22:32

Last Sent: 15th Nov 23 11:22:32

Triggered By: Poonam Bheda

From: contact@embomolmed.org

To: qiaolin.deng@ki.se

Subject: EMM-2023-18672-V3-Q Decision Letter

Message: 15th Nov 2023

Decision on your manuscript EMM-2023-18672-V3-Q

Dear Dr. Deng,

Thank you for your response to the editorial decision on your manuscript entitled "Appropriate glycemic management protects the germline but not uterine environment in type 1". I have now carefully examined the arguments provided in your letter and discussed them with the other members of our editorial team. Additionally, as I previously mentioned to you, I sought external advice on the study from an expert in the field for a second opinion on Reviewer 1's concerns on the model used in your study.

I regret to inform you that we will not be able to reverse our original decision. In line with Reviewer 1, the adviser had significant concerns that the model used in the manuscript is not a good model for human T1D, and rather agree with Reviewer 1 that the low-dose STZ mouse is a model for moderate hyperglycemia. I have included a selection of their comments below, I hope you find them helpful.

I understand that this is disappointing and regret that I could not bring better news this time. I suggest you reconsider the offer to transfer your manuscript LSA.

I hope that this negative decision does not prevent you from considering our journal for the publication of your future studies.

Yours sincerely,

Poonam Bheda

Poonam Bheda, PhD
Scientific Editor
EMBO Molecular Medicine

Advice from Expert:

I cannot disagree with reviewer #1 that the model the authors used cannot be a model of moderate or well managed T1D. T1D is by definition insulin-dependent. An adequate model for this - if the authors want to stick to STZ-treatment - would be insulin-managed overt T1D achieved by a single high-dose STZ injection (as the reviewer also mentioned). Despite the intrinsic problems of the STZ model, what the authors have is a mouse with moderately and constantly high glucose levels (nothing to do with T1D nor with glycemic control). As a reviewer, I would ask the authors to tone down the relevance of their model as a T1D model (in this case there won't be direct relevance for human T1D - which is absent also if they claim their model as a T1D model because in humans this would be a combined glucose/insulin effect). They could instead say they used low dose STZ to generate a model of insulin-sufficient moderate hyperglycaemia.

As a service to authors, EMBO provides authors with the possibility to transfer a manuscript that one journal cannot offer to publish to another EMBO publication. The full manuscript and if applicable, reviewers reports are automatically sent to the receiving journal to allow for fast handling and a prompt decision on your manuscript. For more details of this service, and to transfer your manuscript to another EMBO title please click on *Link Unavailable*

Dear Dr. Deng,

Thank you for the transfer of your manuscript to EMBO reports. I have discussed it and the referee comments with the other members of our team, and I also consulted an expert advisor about it. I am afraid that the outcome of these discussions is not a positive one, as we cannot offer to publish your study in EMBO reports.

We acknowledge that the topic of metabolic state inheritance is of high interest, and that you see no effects on oocytes, which is different compared with the published, related paper using other kinds of STZ injections.

However, the advisor consulted points out that STZ injections cause inflammation, and that the effects on the placenta and fetus can therefore not conclusively be attributed to the hyperglycemia:

"The problem with STZ is the induction of inflammation upon injection which only resolves approx. 10 weeks after injection. So, I feel that the model used is not appropriate to make conclusions about human diabetes. I would ask authors to maintain mice for more than 10 weeks and then start experiments. Also, given all the complications with STZ I feel they have to demonstrate similar results from an alternative model, e.g. NOD."

Given these comments and the concerns raised by the referees, I am sorry to say that we have decided that we cannot offer to publish your manuscript.

I am sorry that I cannot bring better news this time and hope that the experts' comments will be helpful in your continued work in this area.

Kind regards,
Esther

** As a service to authors, EMBO Press provides authors with the ability to transfer a manuscript that one journal cannot offer to publish to another journal, without the author having to upload the manuscript data again. To transfer your manuscript to another EMBO Press journal using this service, please click on Link Not Available

Dear Esther,

Thank you for your email. We are encouraged that you find our manuscript and our results of interest. However, we are quite surprised with the reasoning behind the rejection and would like to express our concerns.

We fully agree with the external advisor that STZ injections cause inflammation, which highlights the need for a maintenance period after injection before the start of any experiment. This is exactly why we have chosen the prolonged time period of at least 10 weeks. This maintenance period is much longer than what any other previous studies to our knowledge so far.

This long maintenance period is also a strength of our study against the conclusion from Chen et al., published in Nature 2022, in which mice were only maintained for 30 days post injection of high dose STZ followed by oocyte collections and subsequent analysis. In their study, we do think general inflammation could be a strong confounding factor. Also, in the field, many other studies start their experiments directly when the STZ-induced mice develop hyperglycemia, which often occur as quickly as 1-2 weeks. In our discussion, we have highlighted this short maintenance period is a strong confounding factor for their follow-up studies. Instead, we have kept much longer for at least 10 weeks with proper phenotyping of these female mice and monitoring the weight etc.

We are also curious about what previous work the external advisor is referring to regarding the post-injection inflammation that resolves 10 weeks after injection. We have not been able to find this reference and would like to kindly ask the reference for this statement, as it is important for our manuscript and the field in general. Even so, we feel like we have answered to the advisor's concern, as we already have maintained our mice for at least 10 weeks before starting any experimental procedures, as stated by the advisor's advice. Therefore, at the time of sacrifice and readout of fetal and placental function and development, at least 12-13 weeks had passed from the last injection.

The external advisor also mentions complications with STZ, but does not specify which ones. The most common complications with STZ induction includes weight loss and renal toxicity, both of which we have managed to exclude in our model (please see our first rebuttal letter as well as our manuscript). Moreover, we have excluded any complications in the reproductive system as our mice have normal oocyte retrieval rates and estrous cyclicity, which otherwise might have affected the fetal and placental development. We therefore believe that we have excluded any confounders associated with STZ-related complications that could affect our results.

Finally, we also have measured Hemoglobin in our mouse model, which confirms that there is no difference between control and STZ-induced mice (see below):

The absence of anemia in STZ-induced mice further validates that our mouse model does not suffer from any long-term inflammation, as this chronic type of inflammation would likely lead to anemia of inflammation.

Taken all together, we believe that our mouse model is appropriate to make conclusions about the effects of hyperglycemia. We apologize that we have not been able to communicate this clearer, and hope that these arguments can provide you with enough confidence to reconsider the decision. I am also very eager to discuss this further with you over phone or zoom.

Best,
Qiaolin

*Qiaolin Deng, PhD, Associate professor
Wallenberg Academy Fellow in Medicine
Dept. Physiology and Pharmacology
Biomedicum B5, Karolinska Institutet
Center for Molecular Medicine
Karolinska University Hospital
Lab website: <http://thedenglab.org>*

Dear Qiaolin,

Thank you for your email. We have decided that we can consider a revised ms that addresses the referee concerns from EMBO Molecular Medicine. We therefore need a fully revised ms and a complete point-by-point response from you. I will contact referees 1 and 2 then with the revised manuscript and we will take it from there.

I would like to suggest that you openly discuss the inflammation issue and that this is the reason that you waited 10 weeks before starting the experiments. I would also suggest that you include the data you have on human oocyte treatment with 10mM glucose. It would be good to discuss all differences between your study compared with the previously published paper by Chen et al. There are, for example, the IVF, the possible impact of inflammation and the different kinds of STZ injections.

Whether the model can be called a T1D model with managed glucose levels or will need to be called a hyperglycemia model will need to be decided by the referees.

Please let me know if you have any questions. You can send us the new files by email, or we can place the ms back in your approval folder, so that you can replace files.

**Karolinska
Institutet**

Dear senior scientific editor of *EMBO Reports*, Dr. Esther Schnapp:

We thank you and the reviewers for constructive comments and for the opportunity to resubmit a revised version of our manuscript. We have addressed all specific questions raised by the reviewers in a complete point-by-point response, please see below. We have also, as suggested by you, openly discussed the issue with acute inflammation associated with STZ injections (lines 408-433 in the revised manuscript), included the data on human oocytes treated with 10mM glucose (lines 274-276, 281-296 and Figure 3b-c in the revised manuscript). We also further discussed all differences between our study and the study performed by Chen et al (lines 435-451 in the revised manuscript). Our changes in the manuscript are highlighted in yellow. We believe that our current manuscript is significantly improved and would be grateful for your consideration.

Yours sincerely,

Qiaolin Deng, PhD, Associate professor
Dept. Physiology and Pharmacology
Karolinska Institutet
Center of Molecular Medicine
Karolinska University Hospital

Referee #1

The two models presented in this study are inadequate to support the claimed conclusions.

The mouse model produced in this study by injecting low doses of streptozotocin (STZ) represents modest cases of type I diabetes (T1D) - not T1D cases whose blood glucose is well controlled by pharmacological treatments. Production of STZ-induced T1D mouse model is a standard practice, and the claim that a low STZ dose produced modest T1D has little novelty.

We thank the reviewer for the comment about our mouse model. We do agree that there is a difference between our T1D model and appropriately managed patients of T1D with treatment. Our model however can offer information on the pathological effects of modestly increased blood glucose levels that are the aim of the current "best practice" in the treatment of pregnant women with diabetes. While previous animal models investigated the effects of extremely increased blood glucose levels, our model is closer to the clinical reality where appropriately managed glycemic levels are achieved, and for which very little information is available in our knowledge.

STZ-induced T1D mouse model is a standard practice for the study of diabetes. However, the critical difference is that our model kept the females largely healthy up to 10 weeks before the evaluation of the oocytes and pregnancy, which minimized any potential STZ-related toxicity. Our study is also the first to carefully phenotype these T1D female mice in major reproductive and metabolic functions. Therefore, the novelty in our experimental design lies in the opportunity to investigate the effect of mild hyperglycemia similar with "state of the art" treatment in pregnant women with diabetes and in minimizing a direct toxic effect of STZ.

We have followed the reviewers' comments and re-formulated our manuscript and incorporated changes to pinpoint that the aim of our study was to investigate the effects of blood glucose levels similar with appropriately managed pregnant women with T1D. Our changes in the manuscript are highlighted in yellow. We believe that these changes have clarified the message of our manuscript.

The IVM culture of human oocytes collected from non-diabetic females in the presence of glucose higher than the standard (5 mM versus 2.5 mM) has little relevance to the T1D patients with controlled glucose level, and the negative outcomes (i.e., lack of morphological or transcriptomic changes) are presented without positive control.

We thank the reviewer for this comment about our experiment investigating the oocyte transcriptome. We do

lack knowledge on the cut-off of glucose levels and their clinical relevance. Therefore, the choice of glucose concentration is guided by the previous study from Chen et al., (Nature, 605, 761-766, 2022) suggesting that performing IVM in 10 or 15mM affects the *TET3* expression in oocytes in contrast to 2.5mM (control conditions) (Figure 1a). We therefore investigated a lower concentration of glucose of 5mM, which is still significantly higher than the control level. Also shown by Chen et al. this glycemic level (5mM) is around the average concentration in the follicular fluid measured in patients with diabetes (Figure 1b).

Figure 1. Excerpt of Extended Data Fig.6 from paper by Chen et al., *Maternal inheritance of glucose intolerance via oocyte *TET3* insufficiency*, Nature, 605, 761-766 (2022). (A) *TET3* mRNA expression in human MII oocytes from IVM under the indicated glucose concentrations. For the 2.5mM, 10mM and 15mM groups, $n=9$, 9 and 11 oocytes, respectively. (B) Human follicular fluid glucose concentrations measured in the patients with or without diabetes. *h*,

We have, in addition, also cultured human oocytes at a higher concentration (10 mM) as a positive control. At this concentration, we still have not observed morphological changes during maturation. However, we do see a significant effect on the oocyte transcriptome, as the number of genes detected and the number of counts per cell are significantly lower for the 10mM group, whilst there is no difference between the 2.5mM and 5mM groups (Figure 2, also in the revised manuscript as Figure 3b-c). We therefore believe that a more severe increase in glycemic levels during IVM will impact the overall oocyte transcriptome, further indicating the importance of proper pregestational control in maternal diabetes. This additional data for the 10mM group has now been included in the manuscript in Figure 3, to clarify that we have validated a positive control. This is described in the manuscript in lines 274-276 as well as 281-296.

Figure 2. Number of counts (UMIs) (a) and genes/features detected (b) per cell for MII oocytes after IVM in 2.5, 5 and 10mM glucose. Significance is tested using one-way ANOVA with Bonferroni's post-hoc test.

Because the clinically very significant claims presented in this manuscript are not supported by adequate models, I think publication of this study, in its current form, may rather mislead the non-expert audience than help their understanding of reproductive impact of T1D.

We agree with the reviewer that it is very difficult to understand the reproductive impact of the T1D using animal models. While traditionally it was investigated using extreme levels of blood glucose in pregnant rodents, our model studies the impact of a glycemic level that is nearly to the "golden standard" of the appropriately managed treatment of pregnant ladies with T1D. Our model offers new data on the effect of another range of blood glucose than previously studied and we believe that it has its own place on the dissection of the mechanisms behind the impact of T1D on reproduction, with it being closer to the clinical reality. We do apologize that we have previously failed to convey the limitation to our model to the reviewer#1 and we have now revised the manuscript accordingly.

Referee #1

In this study, 10-weeks-old female C57BL/6J mice were exposed to low doses of streptozotocin (STZ, 50 mg/kg) by injection - through unspecified route - over five consecutive days. In contrast to the authors' prior experience that the same STZ injection scheme caused severe diabetes in MALE mice, the exposed female mice showed signs of modest type I diabetes (T1D), including "slightly elevated and stable glucose level over the whole experimental period (Fig. 1C)" as well as abnormal OGTT and HbA1c. Different from the commonly used, severe T1D model mice produced by a high-dose and one-shot (150 mg/kg) STZ injection, the modest T1D model presented in this study may have some novelty although the authors do not explicitly state whether the "slight elevation" of blood glucose (Fig. 1C) is statistically significant or not (Fig. 1C does not show any asterisks or ns). The low-STZ T1D model did not affect body weight or estrous cyclicity or transcriptomes of the superovulated oocytes. However, the authors claim that the low-STZ T1D had negative impact to in utero development of fetuses with anomalies observed in placenta.

We thank the reviewer for this comment. We apologize that we have not mentioned the administration route, which is intraperitoneal. We also apologize that the star displaying significance in Fig. 1C has been misplaced in the figure, and instead is above the two data points for week 10. This elevation is significantly different as tested by a two-way ANOVA. Below is the revised Figure 1C.

The authors repeatedly and explicitly emphasize that their low-STZ T1D mouse model resembles clinical human T1D cases whose blood glucose is appropriately managed or controlled. This is my major concern. The low-STZ T1D mouse model presented in this manuscript has never been treated to control their blood glucose level after its initiation of the T1D symptoms, and by mechanism it is a model of modest T1D case with no specific need of treatment. In contrast, human T1D patients whose blood glucose level is high receive pharmacological treatments, and their controlled blood glucose does not mean that their fundamental T1D state (i.e., the lack of endogenous insulin production by the beta cells) is improved. Because of this critical difference between the low-STZ T1D model and the actual T1D patients with controlled blood glucose level, it is misleading to claim that adequate management of blood glucose level prevents oocytes from transcriptional alterations based on the current mouse model. It may be possible to suggest that modest cases of T1D can suffer from fertility issues due to placental anomalies whereas health of their oocytes might be unaffected; however, changing the central claim of this manuscript in such an alternative story does not seem achievable by simply amending the title and text wording.

As we have previously stated we agree with the reviewer that the difference between our mouse model and subjects with T1D with appropriate glycemic control is that the metabolic control is obtained by a different way of insulin provision (endogenous vs exogenous). However, the message we want to convey is that glycemic levels corresponding to present accepted appropriate glycemic management in clinical practice do not affect the oocyte transcriptome or chromatin accessibility but does affect the uterine environment and fetal development. We think our claim is scientifically sound but agree that the limitation of the model in comparison to T1D patients' treatment should be furthermore discussed to clarify any potential misunderstanding. We have therefore clarified this message, and added discussion about this limitation in our manuscript, to highlight the differences between our model and patients with appropriately managed T1D. This discussion is found in lines 479-483.

The authors claim that retrieval of oocytes from T1D patients for research purposes is ethically inappropriate because T1D patients with controlled blood glucose experience low rates of fertility issues. However, even if this claim is acceptable, IVM culture of human oocytes collected from non-diabetic females for IVF purposes in the presence of 2.5 mM or 5 mM glucose needs more justification. Specifically, this experiment lacks

positive control - namely, the authors need to show transcriptomic or morphological changes in the IVM culture oocytes maintained in a higher concentration of glucose. If oocytes are resistant to even very high glucose, then the conclusion will be different from the current manuscript that managed blood glucose of T1D patients may protect oocytes.

This is the same question asked in the Novelty/Model system part by the Reviewer#1. Please also refer to the answer there. In short, we have based our culture condition on previous findings that extreme culture conditions of 10 and 15mM glucose induced transcriptomic changes during IVM, altering *TET3* to potentiate glucose intolerance in offspring. Moreover, as written in the manuscript, we have chosen 5mM as the diabetic condition due to the finding that this is an average level in the follicular fluid of patients with diabetes. We therefore believe that these levels are proper to be tested in IVM. We now also provided new results of 10mM glucose treatment on oocytes as a positive control, in which we see large effects on the overall oocyte transcriptome. These new results are incorporated in the manuscript as mentioned previously.

In the last paragraph of the Results section, the authors referred to Fig. 4h, which I believe an error and needs to be corrected to Fig. 5g.

We apologize for this mistake and have corrected it to Fig. 5g.

The authors discuss that maternal T1D can still affect the uterine environment likely because the placenta is a richly infused organ with blood compared to the ovary. But I do not find adequately strong rationale supporting this speculation.

We thank the reviewer for this comment and have expanded our discussion about this topic in lines 455-460

Referee #2

The study investigated the impact of maternal glycemic control in women with Type 1 Diabetes (T1D) on fetal development and growth outcomes. The findings suggest that even with proper management of glycemic levels, there are still risks associated with T1D pregnancies, such as birth defects, preterm birth, and fetal growth deviations. The study points to the uterine environment as a potentially significant factor in these adverse outcomes, with implications for the future health of the offspring.

The use of a maternal T1D mouse model and the administration of low-dose streptozotocin (STZ) to mimic the condition is an interesting approach. The data showing no changes in the transcriptome of oocytes exposed to managed glucose levels suggests that the impact might not be directly on the oocytes themselves. Instead, the study highlights the role of an adverse uterine environment and placental dysfunction in fetal growth deviations.

The study's emphasis on hypoxia conditions in the placenta and their link to fetal growth restriction is noteworthy. Hypoxia, or inadequate oxygen supply, is known to be a critical factor in various developmental and health outcomes, and understanding the pathways affected by maternal hypoxia in the placenta could provide valuable insights.

It's a valid concern that the study should investigate how hypoxia conditions affect fetal growth and identify specific pathways affected by maternal hypoxia in the fetus. This would help provide a more comprehensive understanding of the mechanisms involved in adverse outcomes in T1D pregnancies and might lead to potential interventions or therapies to mitigate these risks.

Overall, this research appears to be making important contributions to our understanding of diabetic embryopathy and the role of the uterine environment in fetal development, with the suggestion of hypoxia conditions as a critical element. Further investigation into the specific pathways affected by maternal hypoxia could be a valuable next step in this area of study.

We thank the reviewer for these positive remarks.

Referee #3

In this paper, authors aimed to investigate the impact of appropriately managed glycemic levels in maternal type 1 diabetes (T1D) on oocyte transcriptome, chromatin accessibility, intrauterine development, and placental function. The results revealed that appropriately managed maternal glycemic levels preserved the oocyte transcriptome and chromatin accessibility, both in mice and human oocytes. However, fetal growth and placental function were still adversely affected despite glycemic control, highlighting the importance of the uterine environment in developmental programming. Placental dysfunction, characterized by increased

angiogenesis and hypoxia, was identified as a potential contributing factor to fetal growth deviations in the context of appropriately managed maternal T1D. These findings emphasize the significance of achieving proper pregestational glycemic control and the need for further research on therapeutic interventions during pregnancy to mitigate adverse effects on fetal development.

Comments

1) A detailed discussion comparing this study's design and findings with those of Chen et al. could provide valuable insights into the comparison of transcriptomic profiles among control, appropriately managed, and poorly managed glycemia in oocyte transcriptomes.

We thank the reviewer for this comment and agree that further discussion about the study design is important. We have now added a detailed discussion about the differences between our results and the findings of Chen et al., including discussion of other confounding factors and how these can be related to the study design. This can be found in line 435-451 of the revised manuscript.

2) Expanding the study to include human oocytes from type 1 diabetes cases and controls, rather than creating a diabetic environment using 5mM glucose, would enhance the study's relevance. Addressing potential biases arising from comparing transcriptomic profiles of non-diabetic cells cultured in control versus diabetic environments, rather than studying diabetic versus non-diabetic oocytes, should be discussed.

We agree with the reviewer that adding human oocytes from T1D patients would be the best way of investigating the oocyte transcriptome in these patients. However, as described in the manuscript, collection of oocytes from T1D patients solely for research purposes is ethically inappropriate. While the incidence of T1D female subjects is just roughly 0.15% and their need for assisted fertility treatment is not higher than general population, it is also very difficult to obtain these samples in an IVF clinic. Therefore, we chose to use the strategy described in the paper. We do, however, agree that increased discussion of how this model system might not be completely reflective of the in vivo oocyte maturation process is suitable, and this has been added in the discussion part of the manuscript, lines 483-493.

3) The paper should include a discussion of its limitations and potential biases to ensure a more accurate interpretation of the results.

We agree with the reviewer that a section that discuss the limitation and the potential biases is needed for further clarification to strengthen the manuscript. We have therefore added a section in the manuscript about it. This can be found in the manuscript, lines 479-493.

Dear Qiaolin,

Thank you for your patience while your manuscript was re-reviewed at EMBO reports. Referee 1 was unfortunately not available to re-review your manuscript for us, and a new referee 1 was therefore added. While referee 2 agreed to re-review your study, s/he has not sent the report, despite several reminders, and I am therefore making a decision now based on the one referee report we have, in order to save you from further loss of time.

I am happy to say that referee 1 supports the publication of your revised manuscript by EMBO reports. Only a few more minor editorial requests will need to be addressed before we can proceed with the official acceptance of your ms:

- Your manuscript has 5 main figures but is laid out as a full article with separate results and discussion sections. Please either add at least one more main figure or combine the results and discussion sections and reduce the overall character count to 29,000 (excluding references and materials and methods) to publish it as a short report. You can find more info in our guide to authors online.
- Please add up to 5 keywords to the ms file.
- The info under "Code availability" should be moved to the Data Availability section.
- Please rename the conflict of interest subheading to "Disclosure Statement and Competing Interests" and move it to after the Acknowledgments section.
- Please remove all author credits from the ms file. All credits need to be entered online during ms submission.
- The reference format needs to be corrected to the EMBO reports style (in EndNote). It needs to be alphabetical, not numerical, et al needs to be used after 10 author names, year should be in brackets.
- Please send us with your final ms file a completed author checklist that can be found here: <https://www.embopress.org/page/journal/14693178/authorguide>. The completed checklist will also be part of our transparent peer-review file.
- Please enter all funding info also during ms submission into our online system.
- Please remove the blank page from figure 1.
- Please add the missing callout for Fig 3g.
- Please add a title for your supplementary table that should be called Table EV1. Please also correct the callout in the ms text. The title should be added to the excel file.
- Please correct Methods to Materials and Methods.
- Supplementary Figures need to be renamed to Expanded View Figures, e.g. "Figure EV1" instead of "Supplemental figure 1" - all legends, names and callouts need to be updated; especially callouts such as "Supplementary Data Fig. 1" need to be corrected.
- Please remove the old versions of ms files upon submitting your revised manuscript.
- Please add a reviewer access code for the PRJNA1001949 dataset to the data availability section.
- Please address these comments to the figure legends:
 1. Please note that a separate 'Data Information' section is required in the legends of figures 1c-h; 3b-c; 4a, c-d, f, i; 5f-g.
 2. Please indicate the statistical test used for data analysis in the legends of figures 1c-i; 2a, c-d, f; 3b-c, e; 4a, c-d, f, h-i; 5a, c, e-g, supplementary figures 1a-e.
 3. Please note that the box plots need to be defined in terms of minima, maxima, centre, bounds of box and whiskers, and percentile in the legends of figures 2e, h; 3f-g; 5d, supplementary figures 2a-d; 3a; 4a-d.
 4. Please note that information related to n is missing in the legends of figures 2c, f; 3c; 3g; 4a; 5c-d, supplementary figures 1a-e; 2a-d; 3a; 4a-d.
 5. Please note that the error bars are not defined in the legends of figures 1a, c-i; 2a; 4a, c-d, f, h-i; 5a, f-g, supplementary figures 1a-e.
 6. Please note that scale bar and its definition are missing for figures 5a, f-g.
 7. Please note that the white arrows are not defined in the legend of figure 5f. This needs to be rectified.

- The synopsis image at the correct final size of 550 pixels x 380 pixels has blurred text. Can you please send us a new image at the correct size with all text readable? Thank you. Please also send us A) a short (1-2 sentences) summary of the findings and their significance, B) 2-3 bullet points highlighting key results for our journal homepage.

The title is somewhat misleading and needs to be corrected. What about :

Appropriate glycemic management protects the germline but not the uterine environment in hyperglycemia

or

Appropriate glycemic management protects the germline but not the uterine environment in a mouse model of type 1 diabetes

I would also like to suggest a few minor changes to the abstract that needs to be written in present tense. Please let me know whether you agree with the following:

Emerging evidence indicates that parental diseases can impact the health of subsequent generations through epigenetic inheritance. Recently, it was shown that maternal diabetes alters the metaphase II oocyte transcriptome, causing metabolic dysfunction in offspring. However, type 1 diabetes (T1D) mouse models frequently utilized in previous studies may be subject to several confounding factors due to severe hyperglycemia. This limits clinical translatability given improvements in glycemic control for T1D subjects. Here, we optimize a T1D mouse model to investigate the effects of appropriately managed maternal glycemic levels on oocytes and intrauterine development. We show that diabetic mice with appropriate glycemic control exhibit better long-term health, including maintenance of the oocyte transcriptome and chromatin accessibility. We further show that human oocytes undergoing in vitro maturation challenged with mildly increased levels of glucose, reflecting appropriate glycemic management, also retain their transcriptome. However, fetal growth and placental function are affected in mice despite appropriate glycemic control, suggesting the uterine environment rather than the germline as a pathological factor in developmental reprogramming in appropriately managed diabetes.

Referee #1:

The authors have responded appropriately to the reviewer's concerns by adding new data and a critical discussion of limitations of the study. Overall, in its revised version the manuscript now represents an interesting piece of work that may stimulate discussion in the community.

All editorial and formatting issues were resolved by the authors.

Dr. Qiaolin Deng
Karolinska Institutet
Biomedicum B5
Stockholm 17676
Sweden

Dear Qiaolin,

I am very pleased to accept your manuscript for publication in the next available issue of EMBO reports. Thank you for your contribution to our journal.
